# CAN MODEL RANDOMIZATION OFFER ROBUSTNESS AGAINST QUERY-BASED BLACK-BOX ATTACKS?

## ABSTRACT

Deep neural networks are misguided by simple-to-craft, imperceptible adversarial *perturbations* to inputs. Now, it is possible to craft such perturbations solely using model outputs and query-based black-box attack algorithms. These attacks compute adversarial examples by iteratively querying a model and inspecting responses. The attacks' success in near information vacuums poses a significant challenge for developing mitigations. We explore a new idea for a defense driven by a fundamental insight—to compute an adversarial example, the attacks *depend* on the relationship between successive responses to queries to optimize a perturbation. Therefore, to *obfuscate* this relationship, we investigate randomly sampling a model from a set to generate a response to a query. Effectively, this model randomization violates the attacker's expectation of the parameters of a model to remain static between queries to extract information to guide the search toward an adversarial example. It is not *immediately* clear, *if* model randomization can lead to sufficient obfuscation to confuse query-based black-box attacks or *how* best to build such a method. Our *theoretical analysis* proves model randomization always increases resilience to query-based black-box attacks. We demonstrate with extensive empirical studies using 7 state-of-the-art attacks under all *three* perturbation objectives ($l_\infty$, $l_2$, $l_0$) and adaptive attacks, our proposed implementation injects sufficient uncertainty through obfuscation to yield a highly effective defense. Code to be released on GitHub at https://github.com/disco-defense/.

## 1 INTRODUCTION

Many studies comprehensively demonstrate the vulnerability of deep learning models to *adversarial attacks* (Szegedy et al., 2014; Papernot et al., 2017; Carlini & Wagner, 2017; Madry et al., 2018; Athalye et al., 2018). These attacks craft and apply imperceptible perturbations to inputs to mislead or hijack the decision of deep learning models.

In white-box settings, malicious actors can mount strong attacks like Projected Gradient Descent (PGD) with access to model internals and gradient information. However, in practical deployments of machine learning, as with growing numbers of machine learning as a service (MLaaS) offerings, access to model information is highly restricted to external parties. Under these practical settings, an attacker is limited to interacting with a model through a query-response mechanism and only gains access to model outputs. Consequently, in many real-world scenarios, *query-based attacks* in black-box access settings pose the greatest threat. In fact, Ilyas et al. (2018); Guo et al. (2019); Vo et al. (2024) demonstrated practical query-based attacks against models in a real-world system.

Query-based attack algorithms extract response differences to small modifications to the input to estimate gradients or to search for a direction towards an adversarial example. But, the iterative process of making small modifications and estimating gradients or search directions necessitates a myriad of time-consuming interactions with a model (query-responses). This exposes a critical weakness defenders can exploit. The large number of model queries with similar inputs over large time periods is anomalous and raise suspicions. Therefore, a defense objective is to prevent the extraction of useful information from model responses to compute adversarial examples.

**Our Study.** We seek to achieve the objective by injecting uncertainty *directly* into model responses without using random noise.

> *The fundamental insight behind our idea is that computing an adversarial example necessitates successive queries and inspecting responses to make incremental progress towards an adversarial example—the non-source class in Figure 1—but, this progress hinges on the relationship between successive query responses to optimize a perturbation, a process that expects the model parameters to remain static between queries.*

So, we propose randomizing models to obfuscate the relationship between the successive queries and responses to confuse the iterative optimization process. To achieve obfuscation through randomization, we investigate sampling models (or functions) from a set of *diverse* models to respond to each query as illustrated in Figure 1 (last tile). To minimize potential impacts of a defense strategy on performance we investigate learning a set of *well-performing* models.

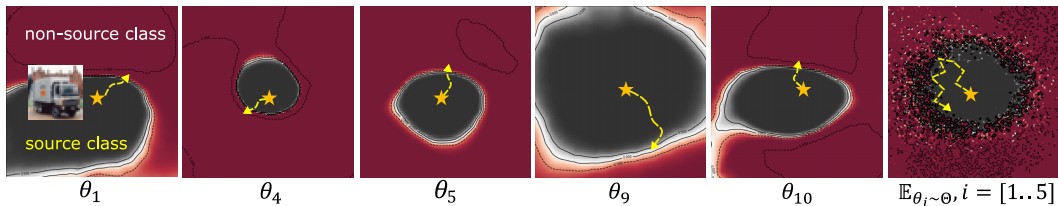

Figure 1: Visualization of decision boundaries for 5 well-performing, diverse models from 10 model parameters ($\theta_1$,..., $\theta_{10}$) and a *randomly* sampled 5 of the 10 for a clean input from the *source class* `Truck` in `CIFAR-10`. Solely using the responses from a single model, whether they be model *decision* labels or *scores*, a query-based attacker can easily estimate gradient directions or search for a path to iteratively move the input towards a decision boundary to build an adversarial example as shown for $\theta_1$, $\theta_4$, $\theta_5$, $\theta_9$ & $\theta_{10}$. But, using responses generated from *randomized models* to mount attacks is much more challenging due to the uncertainty in information derived from such responses. This is especially significant as an input approaches a decision boundary as shown in the *last tile* where each response to a model query is generated from a random sample of five model parameters.

Our theoretical analysis shows the diversity of responses from randomly sampled models can introduce sufficient uncertainty to degrade gradient estimates or misdirect random search attempts. Consequently, building adversarial examples with *score-based* or *decision-based* attack algorithms are made significantly more difficult. Unlike previous methods to confuse attackers, we avoid adding random noise to inputs or features, our thinking mitigates compromising performance for robustness.

Our key contributions can be summarized as follows:

- We investigate the effectiveness of injecting uncertainty into responses by randomized sampling of diverse and well-trained models to respond to queries with a theoretical analysis.

- We implement our idea with techniques for promoting *model diversity* and because we also want random model parameter combinations to be *well performing*, we introduce a *new* learning objective to diversity promotion. The defense framework we investigate is flexible to incorporate other model diversity promotion methods or even existing random noise adding techniques.

- Extensive evaluations with both score-based and decision-based attacks as well as all *three* perturbation objectives ($l_\infty, l_2, l_0$) validate our *theoretical analysis* and demonstrate our method can enhance the robustness against query-based attacks.

## 2 BACKGROUND AND RELATED STUDIES

**Query-based Black-Box Adversarial Attacks.** In contrast to white-box attacks, black-box attackers do not have access to a victim model. One approach is transfer-based black-box attack crafting adversarial examples from a surrogate model to create adversarial examples transferable to a victim model (Papernot et al., 2017; Chen & Liu, 2023). But, transfer-based attacks' success varies significantly due to factors like model hyperparameters, training conditions and constraints in generating

adversarial samples (Chen et al., 2017) and similarity between the surrogates and target models (Suya et al., 2024). In this paper, we focus on defending against query-based adversarial attacks.

Query-based black-box adversarial attacks submit an input to obtain a response from a model. When the response is a confidence score, the attacks operate in a *score-based* threat model; when it is a hard label, a *decision-based* threat model. Two primary approaches to query-based attacks are:

- Gradient estimation methods (Liu et al., 2018a; Ilyas et al., 2018; Cheng et al., 2020; Chen et al., 2020a). In general, these methods estimate the model's gradient with respect to an image $x$ by exploring images surrounding $x$ with queries to assess the model's gradient.
- Gradient-free (search-based) methods (Andriushchenko et al., 2020; Croce et al., 2022; Vo et al., 2022). These methods introduce small random modifications to an image $x$ and observe query response to assess the perturbation's goodness rather than relying on gradient information.

In general, adversarial attacks aim to yield imperceptible perturbations. *Specific* query-based attack algorithm are formulated to minimize three common perturbation objectives $l_2$-norm; $l_\infty$-norm; or $l_0$-norm. In our work, we use *both* score-based and decision-based attacks. And, in contrast to past studies, we evaluate attack algorithms covering all *three* perturbation objectives ($l_\infty, l_2, l_0$).

**Defenses against Black-Box Attacks.** Methods to understand and exploit the anomalous nature of queries attempt to detect attacks (Chen et al., 2020b; Pang et al., 2020; Li et al., 2022). Adversarial training, as a more general method, can be used to defense against adversarial attacks (Tramèr et al., 2018; Zhang et al., 2020; 2022). Similarly, training with noise (Cohen et al., 2019; Salman et al., 2019) can make models robust against adversarial inputs. But, these training approaches can diminish model performance (Zhang et al., 2019; Shafahi et al., 2019; Yang et al., 2020).

By contrast, this paper considers methods to *distort* information available in responses to *deceive* attackers. Intuitively, these methods seek ways to alter the response provided to attackers to misdirect their search towards an adversarial example. To distort information, past work studied: i) adding noise to inputs Cao & Gong (2017); Qin et al. (2021); or ii) injecting noise to model's parameters, activation or adding noise layers (Liu et al., 2018b; He et al., 2019) or iii) adding noise to features (Nguyen et al., 2024). We primarily evaluate the following recent defenses in this paper:

- *Randomize Noise Defense (RND)* Recently, Qin et al. (2021) proposed adding random noise to the input once and theoretically analyzed effectiveness against query-based attacks. Interestingly, Byun et al. (2022) also proposed a randomize noise defense dubbed SND. However, since the two methods are similar, our comparison with RND will extends to both methods.
- *Randomized Features* Nguyen et al. (2024) introduced and explored the idea of adding random noise in feature space.

In contrast, we explore randomization in the function space (effectively the parameter space) to enhance robustness and perhaps do better than noise injections to inputs or features as a defense whilst mitigating the performance impacts of adding noise.

**Adversarial Robustness with Ensemble Diversity.** We consider randomization of functions or models from an ensemble of diverse, well-performing models. Prior studies (Kariyappa & Qureshi, 2019; Pang et al., 2019; Doan et al., 2022) explored ensemble diversity to improve adversarial robustness in *white-box* settings at the cost of sacrificing clean accuracy (Tsipras et al., 2019; Qin et al., 2021). In contrast, we study the potential of ensemble diversity for fortifying models against query-based black-box attacks *without* adversarial training, to mitigate potential performance loss.

A number of diversity promotion methods exist in the literature—we elaborate further in Appendix H.1. In this paper, we evaluate with: i) Deep Ensembles (Ensemble) (Lakshminarayanan et al., 2017; Fort et al., 2020; Wen et al., 2020); ii) DivDis (Lee et al., 2023); iii) DivReg (Teney et al., 2022) and iv) together with a learning objective we propose for Stein Variational Gradient Descent (SVGD) method to learn not only a diverse but also a set of *well-performing* models.

## 3 PROPOSED METHOD

In this section, we formally describe the threat as a problem description, explain our thinking behind our approach for confusing attackers with model randomization, and then provide a theoretical analysis of the convergence of attack algorithms under our our defense method.

### 3.1 PROBLEM FORMULATION

**Score-based Settings.** Given a benign input $\boldsymbol{x} \in \mathbb{R}^d$ and ground truth label $y$, let $f(\boldsymbol{x}, \boldsymbol{\theta})$ denote a victim model with logit score outputs. In untargeted settings, the focus in defense domains, the goal of an adversary is to search for an adversarial example $\tilde{\boldsymbol{x}} \in \mathbb{R}^d$ such that $\arg\max_{\tilde{y}} p(\tilde{y} \mid \tilde{\boldsymbol{x}}) \neq y$ and $\|\boldsymbol{x} - \tilde{\boldsymbol{x}}\|_p \leq B$, where $p(\tilde{y} \mid \tilde{\boldsymbol{x}}) = \text{softmax}\left[f(\tilde{\boldsymbol{x}}; \boldsymbol{\theta})\right]$, $\|.\|_p$ denotes $l_p$ norm and $B$ represents the perturbation budget. Two main approaches to score-based attacks are:

- *Gradient Estimation with Finite Difference Method.* An adversary can estimate the gradient based on the average difference between pairs of output scores returned by the victim model:

$$\tilde{g}_s(\tilde{\boldsymbol{x}}) = \frac{1}{n} \sum_{i=1}^{n} \frac{f(\tilde{\boldsymbol{x}} + \epsilon \boldsymbol{u}_i; \boldsymbol{\theta}) - f(\tilde{\boldsymbol{x}}; \boldsymbol{\theta})}{\epsilon} \boldsymbol{u}_i, \; \boldsymbol{u}_i \sim \mathcal{N}(0, \boldsymbol{I}) \tag{1}$$

- *Gradient-free methods.* An adversary can employ random search or evolutionary algorithms that determine attack direction based on the $f(\tilde{\boldsymbol{x}} + \epsilon \boldsymbol{u}; \boldsymbol{\theta}) - f(\tilde{\boldsymbol{x}}; \boldsymbol{\theta})$. An attack direction is successful if $f(\tilde{\boldsymbol{x}} + \epsilon \boldsymbol{u}; \boldsymbol{\theta}) - f(\tilde{\boldsymbol{x}}; \boldsymbol{\theta}) < 0$.

**Decision-based Settings.** The adversarial objective (untargeted attacks) is to minimize distance $D(\boldsymbol{x}, \tilde{\boldsymbol{x}}) = \|\boldsymbol{x} - \tilde{\boldsymbol{x}}\|_p$ such that $\arg\max_{\tilde{y}} p(\tilde{y} \mid \tilde{\boldsymbol{x}}) \neq y$. Similar to score-based settings, to achieve this objective, an adversary can employ gradient estimation or gradient-free methods. For gradient estimation methods, the gradient can be estimated as follows:

$$\tilde{g}_d(\tilde{\boldsymbol{x}}) = \frac{1}{n} \sum_{i=1}^{n} \frac{D(\boldsymbol{x} + \epsilon \boldsymbol{u}_i, \tilde{\boldsymbol{x}}) - D(\boldsymbol{x}, \tilde{\boldsymbol{x}})}{\epsilon} \boldsymbol{u}_i, \; \boldsymbol{u}_i \sim \mathcal{N}(0, \boldsymbol{I}) \tag{2}$$

### 3.2 CONFUSING ATTACKERS WITH MODEL RANDOMIZATION

Recall that query-based black-box attack algorithms do not have prior knowledge of a target model and are not aware of the defense mechanism employed by the model owner, so they need multiple queries and observations of the model response to estimate a gradient or a search direction. Our fundamental insight is to obfuscate the relationship between query-response pairs. With this in mind, we investigate two ideas we conceptualize in the following hypotheses:

- **Hypothesis 1**. *Randomly selecting a model from a set for responding to a query can obfuscate the relationship between successive pairs of queries and responses.* By employing a *different* function or a learned model to process a query input and generate a response, we can expect to hide the relationship between query-response pairs. Because the attack problem is totally reliant on this relationship, this should lead to sufficient uncertainty to confuse the task of estimating gradient directions or determining search directions towards an adversarial example.
- **Hypothesis 2.** *Enhancing model parameter diversity enhances obfuscation.* Randomly sampling functions or models from a set with very diverse parameters should increase diversity in outputs to further degrade the quality of information extracted from pairs of query-responses.

### 3.3 FORMULATING AND ANALYZING RANDOM MODEL SELECTION

Following on from our *first* hypothesis, we expect feedback from randomly selected model to misdirect gradient and search direction estimation algorithms. However, predicting with an ensemble is shown to lead to higher prediction accuracy (Krogh & Vedelsby, 1994; Dietterich, 2000). Further, the task of building a large number of models to randomly select from can become cumbersome. Consequently, without loss of generality, at each iteration $i$, we consider randomly selecting a subset of models rather than selecting a single model randomly.

Then, given a set of models $\mathcal{F} = \{f(\cdot, \boldsymbol{\theta}_1), f(\cdot, \boldsymbol{\theta}_2), \ldots, f(\cdot, \boldsymbol{\theta}_K)\}$, where $K$ is the total number of models and each model $f(\cdot, \boldsymbol{\theta}_k) \in \mathcal{F}$ with parameters $\boldsymbol{\theta}_k$, the prediction of a subset of models can be formulated as follows:

$$y^* = \arg\max_{y} p(y \mid \boldsymbol{x}), \; \text{where} \; p(y \mid \boldsymbol{x}) = \text{softmax}\left[q(\boldsymbol{x}; \boldsymbol{\pi})\right], \tag{3}$$

where $\boldsymbol{x}$ is the input, $q(\boldsymbol{x}; \boldsymbol{\pi}) = \frac{1}{N} \sum_{k=1}^{N} \pi_k f(\boldsymbol{x}, \boldsymbol{\theta}_k)$, and $\boldsymbol{\pi} \sim \mathcal{B}(\mu_1, \ldots, \mu_K)$ denoting $K$ dimensional vector sampled from $K$ independent Bernoulli distributions and $N$ is the size of the subset of

models. Here, the $\mu_k$ is the mean of a Bernoulli distribution denoting the expected number of times each model is selected.

### 3.3.1 THEORETICAL ANALYSIS OF MODEL RANDOMIZATION AGAINST GRADIENT ESTIMATION ATTACKS

Consider a constant $\epsilon > 0$, $\boldsymbol{u} \sim \mathcal{N}(0, \boldsymbol{I})$, $\boldsymbol{u} \in \mathbb{R}^d$ and $\boldsymbol{x} \in \mathbb{R}^d$, with the output logits of all models expressed as $F(\boldsymbol{x}) = \frac{1}{K} \sum_{k=1}^{K} f_k(\boldsymbol{x}; \boldsymbol{\theta}_k)$ and with a slight misuse of notation, the gradient of such a totality of models can be estimated as follows:

$$\hat{G}(\boldsymbol{x}) = \mathbb{E}_{\boldsymbol{u}} \left[ \frac{F(\boldsymbol{x} + \epsilon\boldsymbol{u}) - F(\boldsymbol{x})}{\epsilon} \boldsymbol{u} \right]; \;\; \hat{G}(\boldsymbol{x}) \in \mathbb{R}^d. \tag{4}$$

Under our model randomization approach described in equation 3, the gradient estimator for a pair of input query samples is:

$$g(\boldsymbol{x}) = \frac{q(\boldsymbol{x} + \epsilon\boldsymbol{u}; \boldsymbol{\pi}^{(2)}) - q(\boldsymbol{x}; \boldsymbol{\pi}^{(1)})}{\epsilon} \boldsymbol{u}; \;\; g(\boldsymbol{x}) \in \mathbb{R}^d. \tag{5}$$

Then, the approximation of the gradient with $n$ different pairs of samples using the finite difference method is formulated as follows:

$$\bar{g}(\boldsymbol{x}) = \frac{1}{n} \sum_{i=1}^{n} \frac{q(\boldsymbol{x} + \epsilon\boldsymbol{u}_i; \boldsymbol{\pi}^{(2i)}) - q(\boldsymbol{x}; \boldsymbol{\pi}^{(2i-1)})}{\epsilon} \boldsymbol{u}_i; \;\; \bar{g}(\boldsymbol{x}) \in \mathbb{R}^d. \tag{6}$$

where defenders generate $\boldsymbol{\pi}^{(2i)} \boldsymbol{\pi}^{(2i-1)} \sim \mathcal{B}(\mu_1, \ldots, \mu_K)$ while attackers generate $\boldsymbol{u}_i \sim \mathcal{N}(0, \boldsymbol{I})$.

**Proposition 1.** *Consider an input $\boldsymbol{x}$ where each element of the gradient $g(\boldsymbol{x})$ estimated at iteration $i$ given in equation 5 is bounded by $a_i^j \leq g(\boldsymbol{x})^j \leq b_i^j$, where $\boldsymbol{a}_i, \boldsymbol{b}_i \in \mathbb{R}^d$, and the average gradient estimator is $\bar{g}(\boldsymbol{x})$ as defined in equation 6. Then, the number of samples $n$ needed such that for every element of $\bar{g}(\boldsymbol{x}) - \hat{G}(\boldsymbol{x})$ is within an error margin $\Delta$ with confidence $1 - \delta$ is at least:*

$$n \geq \sqrt{\frac{\log(\frac{2d}{\delta}) \sum_{i=1}^{n} \left[ \max_j (b_i^j - a_i^j) \right]^2}{2\Delta^2}} \tag{7}$$

*Proof.* We defer the proof to Appendix A ∎

Proposition 1 states that our proposal effectively fortifies against query-based black-box attacks where the cost of the attack, the queries $n$ required to drive $\bar{g}$ closer to $\hat{G}$, is made large to thwart attacks. Additionally, this cost depends on the gradient estimator's bounds, $a_i$ and $b_i$; interestingly, this can be made large when the underlying set of functions or models are able to generate highly diverse outputs to given pairs of inputs.

Importantly, proposition 1 still holds true for *adaptive attacks* applying as we present in Appendix A. Further, the empirical results in Sections 4.1 and 4.2 confirm our observation about the effectiveness and robustness of our defense mechanism against *adaptive attacks*.

### 3.3.2 THEORETICAL ANALYSIS OF MODEL RANDOMIZATION AGAINST GRADIENT-FREE ATTACKS

Consider a constant $\epsilon > 0$ and $\boldsymbol{u} \sim \mathcal{N}(0, \boldsymbol{I})$. Then, the search direction of gradient-free methods against the ensemble of all of the models relies on the sign of $\hat{H}(\boldsymbol{x}, \boldsymbol{u})$ expressed as $\text{sign}(F(\boldsymbol{x} + \epsilon\boldsymbol{u}) - F(\boldsymbol{x}))$, while the search direction of gradient-free methods against randomly selected subsets of models depends on the sign of $\tilde{H}(\boldsymbol{x}, \boldsymbol{u})$ formulated as $\text{sign}(q(\boldsymbol{x} + \epsilon\boldsymbol{u}; \boldsymbol{\pi}^{(i)}) - q(\boldsymbol{x}; \boldsymbol{\pi}^{(j)}))$. As different signs between $\hat{H}(\boldsymbol{x}, \boldsymbol{u})$ and $\tilde{H}(\boldsymbol{x}, \boldsymbol{u})$ or in other words, $\frac{\tilde{H}(\boldsymbol{x}, \boldsymbol{u})}{\hat{H}(\boldsymbol{x}, \boldsymbol{u})} < 0$, represents the mismatch between the attack directions against a random subset of models versus that generated using the entire set of models, $P\left( \frac{\tilde{H}(\boldsymbol{x}, \boldsymbol{u})}{\hat{H}(\boldsymbol{x}, \boldsymbol{u})} < 0 \right)$ represents the probability of misleading an attack direction.

**Proposition 2.** *If we define* $\gamma_{i,j} = q(\boldsymbol{x}; \boldsymbol{\pi}^{(i)}) - q(\boldsymbol{x}; \boldsymbol{\pi}^{(j)})$ *and* $\zeta_i = \nabla q(\boldsymbol{x}; \boldsymbol{\pi}^{(i)}) - \frac{1}{K}\sum_{k=1}^{K} \nabla f_k(\boldsymbol{x}; \boldsymbol{\theta}_k)$, *then the probability of misleading an attack direction, is bounded by:*

$$P\Big(\frac{\tilde{H}(\boldsymbol{x}, \boldsymbol{u})}{\hat{H}(\boldsymbol{x}, \boldsymbol{u})} < 0\Big) \leq \frac{\sqrt{2\mathbb{E}_\pi\Big[\gamma_{i,j}^2 + (\epsilon \boldsymbol{u}\zeta_i)^2\Big]}}{|\hat{H}(\boldsymbol{x}, \boldsymbol{u})|} \tag{8}$$

*Proof.* We defer the proof to Appendix B ∎

We can observe from Proposition 2 that the probability of misleading a search-based attack is low if the model output diversity is low. This is intuitive, since the random selection of diverse models can result in diverse outputs; alternatively, $q(\boldsymbol{x}; \boldsymbol{\pi}^{(i)}) - q(\boldsymbol{x}; \boldsymbol{\pi}^{(j)})$ is positively correlated with the diversity in model outputs.

### 3.4 FORMULATING AND ANALYZING MODEL PARAMETER DIVERSIFICATION

Interestingly, our theoretical analysis of model randomization against gradient estimation and gradient-free attacks already demonstrates promoting model output diversity improves robustness to query based black-box attacks. Then following on from our *second* hypothesis, we investigate learning diverse model parameters for a machine learning task to enhance model output diversity.

In general, we can train an ensemble of models such that their predictions are consistent while their response, such as their output scores are diverse. Formally, the training objective of such a framework can be formulated as follows:

$$\min_{\boldsymbol{\Theta}} \quad \mathbb{E}_{(\boldsymbol{x},y)\sim\mathcal{D}}\Big[\ell\Big(\frac{1}{K}\sum_{k=1}^{K} f(\boldsymbol{x}; \boldsymbol{\theta}_k), y\Big)\Big], \qquad \text{s.t. } \Omega(\mathcal{F}), \tag{9}$$

where $\mathcal{D}$ denotes a training set, $\Omega$ is a set of constraints on the set of functions $\mathcal{F} = \{f(\cdot, \boldsymbol{\theta}_1), f(\cdot, \boldsymbol{\theta}_2), \ldots, f(\cdot, \boldsymbol{\theta}_K)\}$ to ensure diversity is optimized over their parameters $\boldsymbol{\Theta} = \{\boldsymbol{\theta}_1, \boldsymbol{\theta}_2, \ldots, \boldsymbol{\theta}_K\}$ and $\ell(.,.)$ is the loss (*i.e.* cross-entropy). For our defense, there are two pertinent questions that have to be answered in formulation of 9:

- **Question 1:** Because we seek highly diverse models, what constraints encourage the diversity of models leading to high output variance?
- **Question 2:** Since we minimize the loss over the average logits of the set of models, how can we ensure the asymmetry between models promotes high individual model performance because we want any random selection of models to be *well-performing* to minimize the defense strategy's impact on performance?

We discuss our solution in the following sections.

#### 3.4.1 PARAMETER DIVERSITY APPROACH

To answer **Question 1**, we consider the diversity in parameters achieved by adopting an ensemble training framework incorporating a Bayesian formulation of deep learning with Stein variational gradient descent (SVGD) method (Liu & Wang, 2016; Wang & Liu, 2019). This framework allows us to learn a posterior distribution of parameters (weights) and the model parameters sampled from that posterior distribution can result in diverse representations, leading to model diversity.

In this approach, a neural network $f(\mathbf{x}, \boldsymbol{\theta})$ with parameters $\boldsymbol{\theta}$ are considered random variables. Then Bayesian deep learning begins with a prior $p(\boldsymbol{\theta})$ and a likelihood function $p(\mathcal{D}|\boldsymbol{\theta})$ that assesses how well the network with weights $\boldsymbol{\theta}$ fits the data $\mathcal{D}$. The Bayesian inference integrates the likelihood and the prior using Bayes' theorem to derive a *posterior* distribution, $p(\boldsymbol{\theta}|\mathcal{D})$, over the space of weights, given by $p(\boldsymbol{\theta}|\mathcal{D}) = \frac{p(\mathcal{D}|\boldsymbol{\theta})p(\boldsymbol{\theta})}{p(\mathcal{D})}$.

The exact solution for the posterior is often impractical, due to the complexity of deep neural networks and the high-dimensional integral of the denominator, even for networks of moderate size. The true Bayesian posterior distribution is typically a complicated multimodal distribution, making it challenging to accurately sample from. To address these issues, Liu & Wang (2016) proposed Stein Variational Gradient Descent (SVGD) as a general-purpose variational inference algorithm.

Notably, the SVGD method is able to push model parameters apart, directly, it is capable of encouraging learning diversified parameters and provides an effective solution to **Question 1**. Interestingly, the approach is shown to learn different representations Doan et al. (2022) and, consequently, lead to output variance without compromising clean accuracy. Formally, learning diverse parameters, where $\boldsymbol{\theta}_k$ denotes the weights of the $k$-th model, is formulated as follows:

$$\boldsymbol{\theta}_i = \boldsymbol{\theta}_i - \epsilon \phi^*(\boldsymbol{\theta}_i) \text{ and } \phi^*(\boldsymbol{\theta}_i) = \sum_{k=1}^{K} \Big[ \kappa(\boldsymbol{\theta}_k, \boldsymbol{\theta}_i) \nabla_{\boldsymbol{\theta}_i} \ell(f(\boldsymbol{x}; \boldsymbol{\theta}_k), y) - \gamma \nabla_{\boldsymbol{\theta}_i} \kappa(\boldsymbol{\theta}_k, \boldsymbol{\theta}_i) \Big]. \quad (10)$$

Here $\kappa(\cdot, \cdot)$ is a kernel function that encourages model diversity, and $\gamma$ is a hyperparameter to control the trade-off between models' diversity and the loss $\ell(., .)$ (*i.e.* cross-entropy) while $\epsilon$ is the learning rate to determine how much to update each each parameter in each iteration.

Notably, SVGD method was first employed to improve adversarial robustness in white-box settings (Doan et al., 2022). While it incorporates adversarial training, we do not adopt adversarial training, due to the clean accuracy drop. Importantly, the method proposed by Doan et al. (2022) does not consider the problem pertinent to our defence method posed in **Question 2**.

### 3.4.2 NEW TRAINING OBJECTIVE TO ACHIEVE WELL-PERFORMING MODELS

We can observe the training objective in Equation 9 is not able to address the problem posed in **Question 2** as we show in Appendix D. Simply, a naive adoption of the Bayesian training framework with SVGD does not yield models that perform well individually, despite the average performance of all models for a task being high. To address this problem, we propose a new training objective that encourages individual model-learning while training a set of diverse models. We propose incorporating a *sample loss* training objective, $\ell(f(\boldsymbol{x}; \boldsymbol{\theta}_i), y)$, to formulate a new joint loss as follows:

$$\min_{\boldsymbol{\Theta}} \mathbb{E}_{\mathcal{B} \sim \mathcal{D}, \, \boldsymbol{\theta}_i \sim \boldsymbol{\Theta}} \Big[ \mathbb{E}_{(\boldsymbol{x}, y) \sim \mathcal{B}} \Big[ \ell \Big( \frac{1}{K} \sum_{k=1}^{K} f(\boldsymbol{x}; \boldsymbol{\theta}_k), y \Big) + \ell \Big( f(\boldsymbol{x}; \boldsymbol{\theta}_i), y \Big) \Big] \Big], \quad (11)$$

where $\mathcal{B}$ denotes a batch of data sampled from a training set $\mathcal{D}$. Notably, in this training framework, we aim to train all models simultaneously, and for each batch of data $\mathcal{B}$, we uniformly select $\boldsymbol{\theta}_i$ from $\Theta$ at random with replacement.

## 4 EXPERIMENTS AND EVALUATIONS

**Datasets.** We use four different datasets `MNIST` (Lecun et al., 1998), `CIFAR-10` (Krizhevsky et al.), `STL-10` (Coates et al., 2011) and `ImageNet` (Deng et al., 2009).

**Attacks.** We use both *score* and *decision*-based attacks across all three perturbation objectives ($l_\infty$, $l_2$, $l_0$). We emphasize score-based attacks more, as state-of-the-art methods succeed with smaller query budgets. For *score-based* settings under $l_2$, $l_\infty$ and $l_0$ perturbation objectives, we attack with SQUAREATTACK Croce et al. (2022), NESATTACK (Ilyas et al., 2018), SIGNHUNTER (Al-Dujaili & O'Reilly, 2020) and SPARSERS (Andriushchenko et al., 2020). For *decision-based* settings we use HOPSKIPJUMP (Cheng et al., 2019) ($l_2$) and SPAEVO (Vo et al., 2022) ($l_0$).

**Defenses.** Together with ours[1], we compare with randomized input, RND (Qin et al., 2021), and randomized feature, RF (Nguyen et al., 2024), defenses for query-based attacks. Notably, comparing empirical robustness of all adversarial defenses is beyond the scope of this paper. Our aim is to theoretically and empirically examine the effectiveness of a model randomization defense. Nevertheless, for completeness, we compare robustness: **i)** provided by adversarial training (AT) (Wang et al., 2023) in the **Appendix** L; **ii)** AAA (Chen et al., 2022) defending against *only* score-based attacks in **Appendix** M; **iii)** RBC input randomization defense (Cao & Gong, 2017) in **Appendix** N and iv) ADP (Pang et al., 2019) a model diversification approach tested with white-box attacks in **Appendix** O). Additionally, as baselines, we use undefended single models and ensembles where ensemble settings make predictions using all of the models.

**Evaluation Metrics.** Notably, with a defense employing randomness, the same input can result in different outputs (e.g. different scores). An input can also be correctly or incorrectly predicted when repeatedly fed to a defended model adopting randomness. Thus, when an adversarial input created

---

[1]We call ours `Disco`, from the phrase, diversity induced stochastic obfuscation, reflective of our idea.

by an attack aims to fool a model, it could fail or succeed. The more frequently it fails, more robust the randomness defense. Hence, we define the robustness of a randomness-based defense method as follows:

$$\text{Robustness} = \mathbb{E}_{\boldsymbol{x}_{\text{adv}} \sim \mathcal{D}_{\text{ADV}}}[\text{Acc}_{\text{r}}(\boldsymbol{x}_{\text{adv}})], \tag{12}$$

where $\text{Acc}_{\text{r}}(x_{\text{adv}})$ is the number of correct predictions over 1000 predictions of an adversarial example. $\mathcal{D}_{\text{ADV}}$ is a set of adversarial examples generated by an attack.

**Evaluation Protocol.** Recall, when a benign input is fed to a model incorporating randomness, it can be correctly or incorrectly classified. The more frequently a benign input is misclassified, the less reliable the input will be for the purpose of constructing an attack. Although it significantly increases the computation burden, for a fair and reliable comparison, we select benign inputs correctly inferred over *1,000* repeated queries, dubbed *reliable benign inputs*. To manage the computational burden on three different tasks, we compose each evaluation set with 500 reliable benign inputs and use a budget of 10K queries for each attack. For our method, we train a set of 40 models for MNIST task and 10 models for CIFAR-10 and STL-10 tasks and randomly select a subset of 5 models to make predictions. Other selection schemes and results are presented in **Appendix J**.

**Networks & Model Sets.** We use the network in (Cheng et al., 2020) for MNIST, VGG-16 (Liu & Deng, 2015) for CIFAR-10 and ResNet18 (He et al., 2016) for STL-10, then OpenCLIP Radford et al. (2021) for ImageNet. As we discussed in Section 3.3, a more diverse models can improve resilience to attack algorithms. Given our computational constraints and the complexity of different datasets (*i.e.* high dimensional data), we train a larger number of models (40) for the MNIST task and a lower number of models (10) for high-resolution CIFAR-10 and STL-10 with 5 for the ImageNet tasks. Notably, with the ability to relatively quickly learn with a large number of particles (models) with MNIST, we conduct extensive studies using this tasks.

### 4.1 ROBUSTNESS AGAINST QUERY-BASED BLACK-BOX ATTACKS

We report robustness under **7** state-of-the-art attacks, consider all *three* perturbation objectives ($l_2$, $l_\infty$ $l_0$) and include decision and score-based attacks. We evaluate ours and **5** defenses, including adversarial training, with some performance evaluations deferred to the **Appendices** L–O.

*Performance Against Score-Based Attacks.* We report the performance of defense methods against score-based attacks SIGNHUNTER ($l_2$) and SQUAREATTACK ($l_2$) on three different tasks in Table 1. For MNIST, we configure a random selection strategy of five out of 40 models, for CIFAR-10 and STL-10, we configure five out of 10 models. The results in SIGNHUNTER and SQUAREATTACK are strong adversarial attacks. The results demonstrate our method consistently outperforms *state-of-the-art* defenses across tasks and perturbation budgets. This empirical evidence supports our theoretical analysis in Section 3. We provide further evidence, with additional results using different configurations for model randomization in **Appendix J**.

Table 1: $l_2$ **objective.** Robustness (higher ↑ is stronger) of different defense methods against SIGN-HUNTER and SQUAREATTACK.

| MNIST | | | | | | | | | | |
|---|---|---|---|---|---|---|---|---|---|---|
| Methods | SIGNHUNTER | | | | | SQUAREATTACK | | | | |
| | $l_2$ =0.8 | 1.6 | 2.4 | 3.2 | 4.0 | $l_2$ =0.8 | 1.6 | 2.4 | 3.2 | 4.0 |
| Single (*undef*) | 96.8% | 56.6% | 11.0% | 2.0% | 0.0% | 81.4% | 7.0% | 0.0% | 0.0% | 0.0% |
| Ensemble (*undef*) | 98.6% | 93.2% | 56.6% | 13.8% | 3.0% | 95.2% | 38.8% | 0.8% | 0.0% | 0.0% |
| RND | 99.76% | 94.19 % | 78.04% | 63.03% | 49.47% | 99.42% | 92.95% | 77.31% | 60.59% | 45.12% |
| RF | 99.98% | 99.68 % | 96.95% | 86.69% | 70.48% | 99.99% | 99.59% | 95.19% | 83.54% | 68.79% |
| **Disco** | **100%** | **99.99%** | **99.78%** | **99.78%** | **99.25%** | **100%** | **99.76%** | **97.46%** | **88.98%** | **77.75%** |
| CIFAR-10 | | | | | | | | | | |
| | $l_2$ =0.8 | 1.6 | 2.4 | 3.2 | 4.0 | $l_2$ =0.8 | 1.6 | 2.4 | 3.2 | 4.0 |
| Single (*undef*) | 0.2% | 0.0% | 0.0% | 0.0% | 0.0% | 0.2% | 0.0% | 0.0% | 0.0% | 0.0% |
| Ensemble (*undef*) | 15.6% | 1.2% | 0.2% | 0.2% | 0.2% | 7.8% | 0.0 % | 0.0% | 0.0% | 0.0% |
| RND | 99.93% | 98.7 % | 93.75% | 84.1% | 73.81% | 99.03% | 87.49 % | 68.68% | 49.41% | 34.73% |
| RF | **99.98%** | 99.2 % | 94.24% | 85.2% | 75.14% | 99.5% | 91.14 % | 72.34% | 52.31% | 39.59% |
| **Disco** | 99.96% | **99.25%** | **97.61%** | **93.63%** | **90.24%** | **99.56%** | **95.62%** | **87.07%** | **76.5%** | **65.76%** |
| STL-10 | | | | | | | | | | |
| | $l_2$ =1.2 | 2.4 | 3.6 | 4.8 | 6.0 | $l_2$ =1.2 | 2.4 | 3.6 | 4.8 | 6.0 |
| Single (*undef*) | 27.0% | 2.8% | 0.6% | 0.0% | 0.0% | 24.6% | 1.8% | 0.6% | 0.0% | 0.0% |
| Ensemble (*undef*) | 46.2% | 11.4% | 3.0% | 1.0% | 0.6% | 43.0% | 6.6% | 1.0% | 0.4% | 0.2% |
| RND | 99.98% | 99.68 % | 98.92% | 97.19% | 92.74% | 99.93% | 98.63 % | 94.32% | 87.8% | 80.78% |
| RF | 99.99% | 99.63 % | 99.21% | 97.2% | 93.86% | 99.88% | 99.44 % | 97.53% | 95.06% | 89.67% |
| **Disco** | **99.99%** | **99.96%** | **99.8%** | **99.39%** | **98.75%** | **99.97%** | **99.74%** | **98.76%** | **96.86%** | **93.86%** |

Table 2: $l_\infty$ **objective**. Robustness (higher ↑ is stronger) of different defense methods against NESAT-TACK, SIGNHUNTER and SQUAREATTACK with the CIFAR-10 task.

| Attack | Methods | $l_\infty$ =0.02 | 0.04 | 0.06 | 0.08 | 0.1 |
|---|---|---|---|---|---|---|
| NESATTACK | Single (*undef*) | 82.8% | 62.0% | 41.2% | 26.8% | 15.4% |
| | Ensemble (*undef*) | 91.2% | 76.6% | 58.6% | 45.0% | 31.6% |
| | RND | 99.69% | 96.03% | 90.94% | 84.75% | 77.83% |
| | RF | 99.5% | 95.57 % | 88.69% | 85.55% | 79.86% |
| | **Disco** | **99.7%** | **97.93%** | **94.39%** | **90.5%** | **86.77%** |
| SIGNHUNTER | Single (*undef*) | 1.8% | 0.0% | 0.0% | 0.0% | 0.0% |
| | Ensemble (*undef*) | 29.46% | 0.6% | 0.0% | 0.0% | 0.0% |
| | RND | 99.98% | 98.27 % | 88.97% | 75.63% | 63.73% |
| | RF | **99.99%** | 98.51 % | 87.38% | 72.23% | 61.5% |
| | **Disco** | 99.97% | **98.95%** | **95.56%** | **90.7%** | **84.22%** |
| SQUAREATTACK | Single (*undef*) | 2.2% | 0.0% | 0.0% | 0.0% | 0.0% |
| | Ensemble (*undef*) | 28.8% | 1.2% | 0.2% | 0.2% | 0.2% |
| | RND | 99.96% | 90.43 % | 63.68% | 39.44% | 22.06% |
| | RF | 99.92% | 88.97 % | 63.4% | 40.25% | 25.03% |
| | **Disco** | **99.97%** | **96.91%** | **86.52%** | **70.22%** | **55.77%** |

We further evaluate the robustness of defenses against 3 strong, $l_\infty$ attacks, NESATTACK, SIGNHUNTER and SQUAREATTACK as well as $l_0$ attack SPARSERS. The results in Tables 2 and 3 show that our model randomization mechanism is more robust than random noise injection defenses across different attacks and perturbation objectives.

*Performance Against Decision-Based Attacks.* We report results for HOP-SKIPJUMP ($l_2$) and SPAEVO ($l_0$) attacks in Table 4. Our proposed defense demonstrates stronger robustness than the *state-of-the-art* defenses across different decision-based attacks and perturbation objectives. Importantly, the empirical evidence supports our theoretical analysis in Section 3.

### 4.2 ROBUSTNESS AGAINST ADAPTIVE QUERY-BASED BLACK-BOX ATTACKS

We compare RND, RF with our method under *adaptive* SIGNHUNTER and *adaptive* SQUAREATTACK employing Expectation Over Transformation (EOT) (Athalye et al., 2017). Our explanation of EOT-based adaptive attacks is presented in Appendix A. Figure 2 shows that an *adaptive* attacker can alleviate the effect of defense mechanisms

Table 3: $l_0$ **objective**. Robustness (higher ↑ is stronger) of defenses against SPARSERS with CIFAR-10 task.

| Methods | $l_0$ =16px | 32px | 48px | 64px | 80px |
|---|---|---|---|---|---|
| Single (*undef*) | 0.0% | 0.0% | 0.0% | 0.0% | 0.0% |
| Ensemble (*undef*) | 1.6% | 0.0% | 0.0% | 0.0% | 0.0% |
| RND | 45.27% | 23.77% | 15.88% | 11.98% | 8.53% |
| RF | 38.68% | 24.35 % | 20.66% | 15.89% | 14.73% |
| **Disco** | **63.85%** | **47.84%** | **41.59%** | **36.81%** | **31.24%** |

compare to their *non-adaptive* counterparts but with a 10× higher cost for queries. Interestingly, our insights into obfuscating the relationship between query-response pairs with model randomization outperforms random noise injection methods, RND and RF.

Table 4: *Decision-based* **attacks.** Robustness (higher ↑ is stronger) of different defense methods against HOPSKIPJUMP ($l_2$) and SPAEVO ($l_0$) with the CIFAR-10 task.

| Methods | HOPSKIPJUMP | | | | | SPAEVO | | | | |
|---|---|---|---|---|---|---|---|---|---|---|
| | $l_2$ =0.8 | 1.6 | 2.4 | 3.2 | 4.0 | $l_0$ =4px | 8px | 12px | 16px | 20px |
| Single (*undef*) | 0.0% | 0.0% | 0.0% | 0.0% | 0.0% | 43.2% | 19.6% | 7.2% | 3.8% | 3.0% |
| Ensemble (*undef*) | 3.2% | 0.0% | 0.2% | 0.2% | 0.2% | 59.2% | 29.6% | 17.2% | 8.4% | 6.4% |
| RND | 99.94% | 99.13 % | 98.23% | 96.71% | 95.95% | 93.45% | 93.08% | 92.86% | 92.83% | 92.82% |
| RF | 99.91% | 98.53 % | 97.13% | 95.57% | 94.34% | 91.84% | 91.49% | 91.4% | 91.4% | 91.4% |
| **Disco** | **99.94%** | **99.4%** | **98.77%** | **98.04%** | **97.43%** | **96.17%** | **95.99%** | **95.94%** | **95.88%** | **95.84%** |

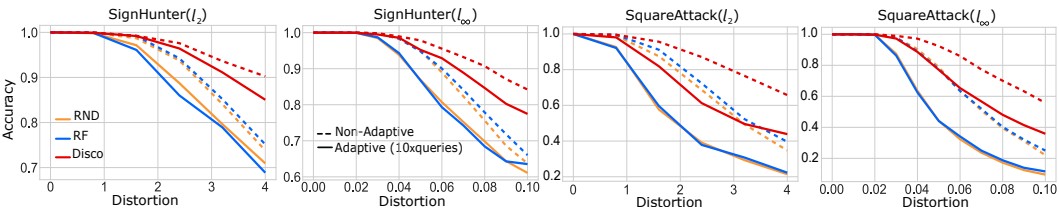

Figure 2: Robustness against *non-adaptive* vs. *adaptive* $l_2$ and $l_\infty$ attacks using SIGNHUNTER and SQUAREATTACK. For *adaptive* attacks, the adversary expends extra, $m = 10\times$ queries for each input sample, and averages the outputs to mitigate our obfuscation (we defer details to Appendix A).

### 4.3 INVESTIGATING CLEAN ACCURACY OF UNDEFENDED AND DEFENDED MODELS

Defenses invariably compromise clean accuracy for robustness. We report clean accuracy achieved by undefended and defended models along with the resulting clean accuracy drop (CAD) denoted by (↓ $\Delta$) across the tasks in Tables 5. Importantly, our goal to seek *well-performing* models with the

incorporation of the new learning objective in Section 3.4.2 has mitigated the CAD drop significantly better than state-of-the-art random noise defenses. We provide further evidence to demonstrate the impact of the learning objective in **Appendix** D.

Table 5: Clean accuracy achieved by undefended and defended models. For our method, SVGD+ (All) is a clean accuracy of the entire model set while Disco(SVGD+) presents the clean accuracy of random five out of 40 models (MNIST) or random five out of 10 models (CIFAR-10, STL-10)— we report clean accuracy for other model randomization configurations in **Appendix** J.3.

| Dataset | Baselines | | Defense Methods | | Ours | |
|---|---|---|---|---|---|---|
| | Single Model | Ensembles | RND ($\downarrow \Delta$) | RF ($\downarrow \Delta$) | SVGD+ (All) | Disco(SVGD+) ($\downarrow \Delta$) |
| MNIST | 99.64% | 99.72% | 98.59% ($\downarrow$1.05%) | 98.45% ($\downarrow$1.19%) | 99.59% | 99.34% ($\downarrow$**0.25%**) |
| CIFAR-10 | 92.09% | 94.76% | 87.63% ($\downarrow$4.46%) | 89.73% ($\downarrow$2.36%) | 93.19% | 92.26% ($\downarrow$**0.93%**) |
| STL-10 | 90.39% | 92.15% | 86.38% ($\downarrow$4.01%) | 88.5% ($\downarrow$1.89%) | 90.18% | 88.97% ($\downarrow$**1.21%**) |

### 4.4 RELATIONSHIP BETWEEN ROBUSTNESS AND DIVERSITY PROMOTING ALTERNATIVES

We assess alternative methods for promoting model diversity (Deep Ensembles, Ensemble, (Lakshminarayanan et al., 2017); DivDis (Lee et al., 2023); DivReg (Teney et al., 2022)) to: i) evaluate their performance under our model randomization method; and ii) understand the relationship between robustness and model diversity (we defer formulations of these training objectives to **Appendix** H.1).

*Performance.* We compare the robustness of alternatives with SVGD+ under our framework against score-based, $l_2$ adversarial attacks SIGN-HUNTER and SQUAREATTACK. The results in Table 6 show that our method outperforms the alternatives. We report additional results against other attacks in **Appendix** I.

*Model Diversity Analysis.* We discussed in Section 3.4 how greater diversity among models can enhance obfuscation and lead to enhanced robustness against query-based attacks. Here, we use Jensen–Shannon (JS) divergence to understand model diversity promoted by different methods. While JS divergence and additional results are detailed in **Appendix** H.2, here, Figure 3 shows our proposed method (SVGD+) to achieve greater diversity than the alternatives; importantly, this pro-

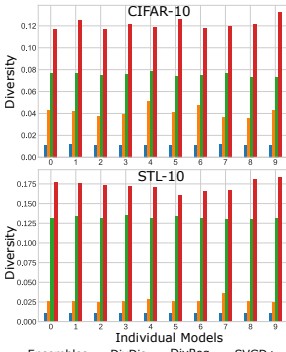

Figure 3: Model diversity using JS divergence among Ensemble, DivDis, DivReg and SVGD+ (**ours**).

vides evidence to support our second hypothesis in Section 3.2 and provides further insights to explain the better robustness achieved with Disco implemented with SVGD+.

Table 6: $l_2$ **objective attacks, CIFAR-10.** Robustness (higher $\uparrow$ is stronger) of diversity promotion approaches against SIGNHUNTER and SQUAREATTACK with the CIFAR-10 task.

| Methods | SIGNHUNTER | | | | | SQUAREATTACK | | | | |
|---|---|---|---|---|---|---|---|---|---|---|
| | $l_2$ =0.8 | 1.6 | 2.4 | 3.2 | 4.0 | $l_2$ =0.8 | 1.6 | 2.4 | 3.2 | 4.0 |
| Disco(Ensemble) | 99.69% | 97.72% | 93.49% | 89.22% | 81.55% | 98.2% | 87.9 % | 76.9% | 63.5% | 52.2% |
| Disco(DivDis) | 99.87% | 98.74 % | 95.32% | 91.97% | 85.6% | 99.1% | 94.1 % | 82.7% | 70.4% | 56.4% |
| Disco(DivReg) | 99.96% | 99.07% | 96.38% | 91.42% | 86.95% | 99.2% | 90.9 % | 76.3% | 64.6% | 51.6% |
| **Disco(SVGD+)** | **99.96%** | **99.25%** | **97.61%** | **93.63%** | **90.24%** | **99.56%** | **95.62%** | **87.07%** | **76.5%** | **65.76%** |

### 4.5 COST ANALYSIS AND AMELIORATING COSTS TO DEFEND IMAGENET ON OPENCLIP

We analyses the cost overhead of Disco in Appendix E. In Appendix F we adopt the method of model fine tuning to implement Disco for a *practical task*, represented by ImageNet, and *for a practical, large-scale model*, represented by OpenCLIP—now the cost burden is < 1%.

## 5 CONCLUSION

This study investigates the effectiveness of a model randomization defense against query-based black-box attacks in both score-based and decision-based settings. We theoretically analyze the defense and prove the link between diversity of model outputs to model robustness; hence, model randomization always increases resilience to query-based black-box attacks. We realize the approach by learning a set of diverse yet well performing models for random selection to provide robustness whilst minimizing the clean accuracy drop of defended models. We demonstrate the approach leads to an effective defense with 7 state-of-the-art query-based black-box attacks under all *three* perturbation objectives ($l_\infty, l_2, l_0$).

ACKNOWLEDGMENTS

Anonymous for review.

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

OVERVIEW OF MATERIALS IN THE APPENDIX

We provide a brief overview of the additional experimental results and findings in the Appendices that follow.

1. Proofs for theoretical analysis against gradient estimation attacks (Appendix A) and gradient-free attacks (Appendix B).

2. Analysis of Trade-off Between Subset Set Size and Error Estimation C.

3. Effectiveness of the proposed learning objective in Section 3.4.2 (Appendix D).

4. Cost analysis of Disco method (Appendix E)

5. Cost mitigation strategy and effectiveness on a high-resolution large-scale dataset with a practical large-scale OpenCLIP model (Appendix F).

6. Effectiveness against an attack in a surrogate model setting (Appendix G).

7. Formulations of and diversity analysis of alternative approaches for promoting model diversity (Appendix H).

8. Robustness evaluations of alternative approaches for model diversity promotion against **4** state-of-the-art attacks under $l_\infty$ and $l_0$ perturbation objectives (Appendix I).

9. Robustness and clean accuracy evaluations of different randomized model selection strategies with different model diversification methods (Appendix J).

10. Robustness over multiple trials (Monte Carlo experiment) (Appendix K).

11. Robustness comparisons with **3** additional defenses:

   - Adversarial Training (AT) defense (Appendix L).
   - Adversarial Attack on Attackers (AAA) Defense (Appendix M).
   - Region-Based Classification (RBC) (Appendix N).

12. Robustness comparison with Adaptive Diversity Promoting (ADP) method (Appendix O).

# A    PROOF FOR THEORETICAL ANALYSIS AGAINST GRADIENT ESTIMATION ATTACKS

In this section, we provide the theoretical analysis of our defense method against gradient-estimation attacks and the proof of proposition 1.

*Proof.* We consider gradient estimation when the entire set of models (or even a single model) is presented to the attacker versus the expectation of gradient estimation under different subsets.

Given an input $\boldsymbol{x} \in \mathbb{R}^d$ and $K$ models, the output logits of $K$ models is $F(\boldsymbol{x}) = \frac{1}{K} \sum_{k=1}^{K} f(\boldsymbol{x}; \boldsymbol{\theta}_k)$, where $\mathcal{F} = \{f(\cdot, \boldsymbol{\theta}_1), f(\cdot, \boldsymbol{\theta}_2), \dots, f(\cdot, \boldsymbol{\theta}_K)\}$. Given $\epsilon > 0$, $\boldsymbol{u} \sim \mathcal{N}(0, \boldsymbol{I})$, $\boldsymbol{u} \in \mathbb{R}^d$, the *gradient estimated* when the entire model set is used can be formulated as follows:

$$\hat{G}(\boldsymbol{x}) = \mathbb{E}_{\boldsymbol{u}} \left[ \frac{F(\boldsymbol{x} + \epsilon \boldsymbol{u}) - F(\boldsymbol{x})}{\epsilon} \boldsymbol{u} \right]; \ \ \hat{G}(\boldsymbol{x}) \in \mathbb{R}^d.$$

Applying Taylor expansion at $\boldsymbol{x}$, we have $F(\boldsymbol{x} + \epsilon \boldsymbol{u}) \approx F(\boldsymbol{x}) + \epsilon \boldsymbol{u} \nabla F(\boldsymbol{x})$. Then, we have:

$$\hat{G}(\boldsymbol{x}) \approx \mathbb{E}_{\boldsymbol{u}} \left[ \frac{F(\boldsymbol{x}) + \epsilon \boldsymbol{u} \nabla F(\boldsymbol{x}) - F(\boldsymbol{x})}{\epsilon} \boldsymbol{u} \right] \approx \mathbb{E}_{\boldsymbol{u}}[\boldsymbol{u} \nabla F(\boldsymbol{x}) \boldsymbol{u}]$$

$$\approx \mathbb{E}_{\boldsymbol{u}} \left[ \boldsymbol{u} \frac{1}{K} \Big[ \sum_{k=1}^{K} \nabla f(\boldsymbol{x}, \boldsymbol{\theta}_k) \Big] \boldsymbol{u} \right]$$

Notably, since $\boldsymbol{u}$ is sampled from a normal distribution, it provides an unbiased estimation of the gradient.

However, under our defense, a subset of models is sampled uniformly at random with replacement from $\mathcal{F}$ and an average is taken to make a prediction. Particularly, we sample $q(\boldsymbol{x}; \boldsymbol{\pi}) = \frac{1}{N} \sum_{k=1}^{N} \pi_k f(\boldsymbol{x}, \boldsymbol{\theta}_k)$ where $\boldsymbol{\pi} \sim \mathcal{B}(\mu_1, \dots, \mu_K)$ denotes a $K$ dimensional vector sampled from $K$ independent Bernoulli distributions and $N$ is the size of the model subset. Thus, the expectation of the estimated gradient from all subsets of models can be formulated as the following:

$$\tilde{G}(\boldsymbol{x}) = \mathbb{E}_{\boldsymbol{u}} \left[ \mathbb{E}_{\pi} \left[ \frac{q(\boldsymbol{x} + \epsilon \boldsymbol{u}; \boldsymbol{\pi}^{(i)}) - q(\boldsymbol{x}; \boldsymbol{\pi}^{(j)})}{\epsilon} \right] \boldsymbol{u} \right], \ \ \tilde{G}(\boldsymbol{x}) \in \mathbb{R}^d.$$

where $i, j$ denotes $i$- and $j-$th consecutive iterations (model queries). Applying Taylor expansion at $\boldsymbol{x}$, we have $q(\boldsymbol{x} + \epsilon \boldsymbol{u}; \boldsymbol{\pi}^{(i)}) \approx q(\boldsymbol{x}; \boldsymbol{\pi}^{(i)}) + \epsilon \boldsymbol{u} \nabla q(\boldsymbol{x}; \boldsymbol{\pi}^{(i)})$. Then, we have:

$$\tilde{G}(\boldsymbol{x}) \approx \mathbb{E}_{\boldsymbol{u}} \left[ \mathbb{E}_{\pi} \left[ \frac{q(\boldsymbol{x}; \boldsymbol{\pi}^{(i)}) + \epsilon \boldsymbol{u} \nabla q(\boldsymbol{x}; \boldsymbol{\pi}^{(i)})] - q(\boldsymbol{x}; \boldsymbol{\pi}^{(j)})}{\epsilon} \right] \boldsymbol{u} \right]$$

$$\approx \mathbb{E}_{\boldsymbol{u}} \left[ \mathbb{E}_{\pi} \left[ \frac{q(\boldsymbol{x}; \boldsymbol{\pi}^{(i)}) - q(\boldsymbol{x}; \boldsymbol{\pi}^{(j)}) + \epsilon \boldsymbol{u} \nabla q(\boldsymbol{x}; \boldsymbol{\pi}^{(i)})}{\epsilon} \right] \boldsymbol{u} \right]$$

$$\approx \mathbb{E}_{\boldsymbol{u}} \left[ \mathbb{E}_{\pi} \left[ \frac{q(\boldsymbol{x}; \boldsymbol{\pi}^{(i)}) - q(\boldsymbol{x}; \boldsymbol{\pi}^{(j)})}{\epsilon} + \boldsymbol{u} \nabla q(\boldsymbol{x}; \boldsymbol{\pi}^{(i)}) \right] \boldsymbol{u} \right]$$

$$\approx \mathbb{E}_{\boldsymbol{u}} \left[ \left[ \frac{1}{\epsilon} \Big[ \underbrace{\mathbb{E}_{\pi} \Big[ q(\boldsymbol{x}; \boldsymbol{\pi}^{(i)}) \Big]}_{\text{A}} - \underbrace{\mathbb{E}_{\pi} \Big[ q(\boldsymbol{x}; \boldsymbol{\pi}^{(j)}) \Big]}_{\text{B}} \Big] + \boldsymbol{u} \mathbb{E}_{\pi} \Big[ \nabla q(\boldsymbol{x}; \boldsymbol{\pi}^{(i)}) \Big] \right] \boldsymbol{u} \right]$$

Since:

$$\mathbb{E}_{\pi} \Big[ q(\boldsymbol{x}; \boldsymbol{\pi}^{(i)}) \Big] = \sum_{k}^{K} \mu_k f(\boldsymbol{x}; \boldsymbol{\theta}_k),$$

and the difference between the first two expectation terms A and B will approach zero, we have:

$$\tilde{G}(\boldsymbol{x}) \approx \mathbb{E}_{\boldsymbol{u}} \left[ \boldsymbol{u} \mathbb{E}_{\pi} \Big[ \nabla q(\boldsymbol{x}; \boldsymbol{\pi}^{(i)}) \Big] \boldsymbol{u} \right] \approx \mathbb{E}_{\boldsymbol{u}} \left[ \boldsymbol{u} \Big[ \sum_{k}^{K} \mu_k \nabla f(\boldsymbol{x}; \boldsymbol{\theta}_k) \Big] \boldsymbol{u} \right]$$

Thus, we have:

$$\tilde{G}(\boldsymbol{x}) - \hat{G}(\boldsymbol{x}) \approx \mathbb{E}_{\boldsymbol{u}}\left[\boldsymbol{u}\left[\sum_{k=1}^{K}(\mu_k - \frac{1}{K})\nabla f(\boldsymbol{x}; \boldsymbol{\theta}_k)\right]\boldsymbol{u}\right].$$

As $\mu_k$ approaches $\frac{1}{K}$, the difference between $\tilde{G}(\boldsymbol{x})$ and $\hat{G}(\boldsymbol{x})$ approaches zero.

*Given the result above, first we consider non-adaptive attackers.*

**Non-adaptive attack setting.** Generally, an adversary does not have knowledge of defense mechanisms. Hence, under our defense mechanism, the gradient estimator with a pair of samples is:

$$g(\boldsymbol{x}) = \frac{q(\boldsymbol{x} + \epsilon\boldsymbol{u}; \boldsymbol{\pi}^{(2)}) - q(\boldsymbol{x}; \boldsymbol{\pi}^{(1)})}{\epsilon}\boldsymbol{u}, \;\; g(\boldsymbol{x}) \in \mathbb{R}^d.$$

In practice, to achieve a good approximation of gradient, the finite difference method samples multiple $\boldsymbol{u}$, obtains multiple $g(\boldsymbol{x})$ and takes the average. Then, the approximation of the gradient with $n$ different pairs of samples using the finite difference method is formulated as follows:

$$\bar{g}(\boldsymbol{x}) = \frac{1}{n}\sum_{i=1}^{n}\frac{q(\boldsymbol{x} + \epsilon\boldsymbol{u}_i; \boldsymbol{\pi}^{(2i)}) - q(\boldsymbol{x}; \boldsymbol{\pi}^{(2i-1)})}{\epsilon}\boldsymbol{u}_i, \;\; \bar{g}(\boldsymbol{x}) \in \mathbb{R}^d.$$

where the defender generates $\boldsymbol{\pi}^{(2i)}, \boldsymbol{\pi}^{(2i-1)} \sim \mathcal{B}(\mu_1, \ldots, \mu_K)$, the attacker generates $\boldsymbol{u}_i \sim \mathcal{N}(\boldsymbol{0}, \boldsymbol{I})$. However, there is a gap between this approximation $\bar{g}(\boldsymbol{x})$ and the *expected gradient estimation* of all subsets $\tilde{G}(\boldsymbol{x}) = \mathbb{E}[g(\boldsymbol{x})]$. Since we proved the *expected gradient estimation* $\tilde{G}(\boldsymbol{x})$ approximates the *actual gradient estimation* of the entire model set $\hat{G}(\boldsymbol{x})$, $|\bar{g}(\boldsymbol{x}) - \tilde{G}(\boldsymbol{x})|$ approximates to $|\bar{g}(\boldsymbol{x}) - \hat{G}(\boldsymbol{x})|$. If each element of the gradient $g(\boldsymbol{x})$ estimated at iteration $i$ is bounded by $a_i^j \leq g(\boldsymbol{x})^j \leq b_i^j$ with $\boldsymbol{a}_i, \boldsymbol{b}_i \in \mathbb{R}^d$, $n$ different gradient estimators $g(.)$ are independent random variables and $A^j$ is defined as $|\bar{g}(x)^j - \hat{G}(x)^j| \geq \Delta$, according to the Hoeffding's inequality and employing a union bound over all $d$ dimensions to bound the probability of deviation in any component, we have:

$$P(\cup_{j=1}^{d}A^j) \leq \sum_{j=1}^{d}P(A^j) = \sum_{j=1}^{d}2\exp\left(-\frac{2n^2\Delta^2}{\sum_{i=1}^{n}(a_i^j - b_i^j)^2}\right).$$

Where $\Delta$ is a gap or margin of error. This term can further be upper bounded by considering the fact that $\exp(-x)$ is monotonically decreasing, we know for any $j$:

$$\exp\left(-\frac{2n^2\Delta^2}{\sum_{i=1}^{n}(a_i^j - b_i^j)^2}\right) \leq \exp\left(-\frac{2n^2\Delta^2}{\sum_{i=1}^{n}[\max_j(a_i^j - b_i^j)^2]}\right)$$

Therefore, we have:

$$P(\cup_{j=1}^{d}A^j) \leq \sum_{j=1}^{d}2\exp\left(-\frac{2n^2\Delta^2}{\sum_{i=1}^{n}(a_i^j - b_i^j)^2}\right) \leq 2d\exp\left(-\frac{2n^2\Delta^2}{\sum_{i=1}^{n}[\max_j(b_i^j - a_i^j)]^2}\right)$$

To achieve low margin of error $\Delta$, the upper bound of the probability such that this gap is beyond $\Delta$ must be low. To achieve this with the desired confidence level $1 - \delta$ and the given bound as above, we set the right-hand side of the inequality smaller than $\delta$ and solve for $n$ as the following:

$$2d\exp\left(-\frac{2n^2\Delta^2}{\sum_{i=1}^{n}[\max_j(b_i^j - a_i^j)]^2}\right) \leq \delta$$

$$-\frac{2n^2\Delta^2}{\sum_{i=1}^{n}[\max_j(b_i^j - a_i^j)]^2} \leq \log\frac{\delta}{2d}$$

$$\frac{2n^2\Delta^2}{\sum_{i=1}^{n}[\max_j(b_i^j - a_i^j)]^2} \geq \log\frac{2d}{\delta}$$

$$n^2 \geq \frac{\log\frac{2d}{\delta}\sum_{i=1}^{n}[\max_j(b_i^j - a_i^j)]^2}{2\Delta^2}$$

$$n \geq \sqrt{\frac{\log\frac{2d}{\delta}\sum_{i=1}^{n}[\max_j(b_i^j - a_i^j)]^2}{2\Delta^2}}$$

This implies that when a set of models is more diverse, the bound $a_i^j < g(\boldsymbol{x})^j < b_i^j$ is larger, and the number of samples $n$ needed, such that every element of $\bar{g}(\boldsymbol{x}) - \hat{G}(\boldsymbol{x})$ is more likely within the error margin $\Delta$, grows significantly. ∎

*Next we consider adaptive attackers.*

**Adaptive attack setting.** Now, we assume the attacker has knowledge of the defense mechanism and is aware that a subset of $K$ models is randomly selected to generate the response to a model query. If an adversary has prior knowledge of our defense mechanism, they can employ Expectation Over Transformation (EOT) Athalye et al. (2017) to obtain a more accurate gradient estimate.

It is worth noting that, in the original EOT method, the adversarial perturbation gradient is calculated based on a series of transformed inputs (adversarial examples). The reason is to address the issue incurred by the ineffectiveness of an adversarial example yielded by an adversary when the adversarial example is randomly transformed (Athalye et al., 2017) *i.e.* with different view angles. Therefore, to maintain the effectiveness of an adversarial example over different transformations, they model these transformations in their optimization procedure by transforming the inputs. Similarly, to maintain the effectiveness of an adversarial example over different models whose selection is represented by different $\boldsymbol{\pi}$, an adversary seeks a perturbation gradient direction such that it is effective over different models. Therefore, in our study, the so-called EOT is performed over $\boldsymbol{\pi}$. Interestingly, the same reasoning is in Athalye et al. (2018) and Nguyen et al. (2024) when applying EOT to attack defenses involving stochasticity, like with ours.

In practice, similar to a non-adaptive attack, an EOT-based adaptive attack sends $m$ queries to a target model to estimate the gradient, but for each query, it feeds a target model with the same input $n$ times to mitigate the impact of randomness. As a result, the number of queries to estimate a gradient in the adaptive setting is $m \times n$. This is $m\times$ higher than a non-adaptive attack. For instance, if a non-adaptive attack uses 10K queries and $m = 10$, the total number of queries needed by an adaptive attacker is 100K. Likewise, for gradient-free attacks, each input is fed into a target model $m$ times to find a more reliable attack direction.

Formally, under our defense mechanism, the gradient estimator employed by an adaptive attack using the finite difference method is formulated as follows:

$$g'(\boldsymbol{x}) = \frac{1}{m} \sum_1^m \frac{f^{(i)}(\boldsymbol{x} + \epsilon\boldsymbol{u}; \boldsymbol{\pi}^{(i)}) - f^{(j)}(\boldsymbol{x}; \boldsymbol{\pi}^{(j)})}{\epsilon} \boldsymbol{u} = \frac{1}{m} \sum_1^m g(\boldsymbol{x})$$

where $\boldsymbol{\pi}^{(i)}, \boldsymbol{\pi}^{(j)} \sim \mathcal{B}(\mu_1, \ldots, \mu_K)$ is generated by the defender, $\boldsymbol{u} \sim \mathcal{N}(0, \boldsymbol{I})$ is generated by the attacker and $f^{(i)}(\boldsymbol{x}; \boldsymbol{\pi}^{(i)}) = \sum_k \boldsymbol{\pi}_k^{(i)} f_k(\boldsymbol{x}, \boldsymbol{\theta}_k)$, $f^{(j)}(\boldsymbol{x}; \boldsymbol{\pi}^{(j)}) = \sum_k \boldsymbol{\pi}_k^{(j)} f_k(\boldsymbol{x}, \boldsymbol{\theta}_k)$.

Similar to the non-adaptive setting, to achieve a good approximation of gradient, the finite difference method samples multiple $\boldsymbol{u}$, obtains multiple $g'(\boldsymbol{x})$ and takes the average $\tilde{g}(\boldsymbol{x}) = \frac{1}{n} \sum_1^n g'(\boldsymbol{x})$. Then, we have $\tilde{g}(\boldsymbol{x}) = \frac{1}{n'} \sum_1^n \sum_1^m g(\boldsymbol{x})$ with $n' = n \times m$. If each element of the gradient $g'(\boldsymbol{x})$ estimated at iteration $i$ is bounded by $a_i'^j \leq g'(\boldsymbol{x})^j \leq b_i'^j$ with $\boldsymbol{a}_i', \boldsymbol{b}_i' \in \mathbb{R}^d$, and $A'^j$ is defined as $|\tilde{g}(x)^j - \hat{G}(x)^j| \geq \Delta$, according to the Hoeffding's inequality and employing a union bound over all $d$ dimensions to bound the probability of deviation in any component, we have:

$$P(\cup_{j=1}^d A'^j) \leq \sum_{j=1}^d P(A'^j) = \sum_{j=1}^d 2 \exp\Big(-\frac{2n'^2\Delta^2}{\sum_{i=1}^n (a_i'^j - b_i'^j)^2}\Big)$$

The number of samples $n'$ needed to ensure every element of $\tilde{g}(\boldsymbol{x}) - \hat{G}(\boldsymbol{x})$ more likely within an error margin $\Delta$ with confidence $1 - \delta$ is at least:

$$n' \geq \sqrt{\frac{\log \frac{2d}{\delta} \sum_{i=1}^n [\max_j (b_i'^j - a_i'^j)]^2}{2\Delta^2}}$$

This implies that the number of samples $n'$ relies on the range of estimator $d'$ with a given confidence interval and margin of error. Importantly, as similar to non-adaptive attacks, when a set of models is more diverse, the bound $a_i'^j \leq g'(\boldsymbol{x})^j \leq b_i'^j$ is larger, and the number of samples $n'$ needed, such that every element of $\tilde{g}(\boldsymbol{x}) - \hat{G}(\boldsymbol{x})$ is more likely within the error margin $\Delta$, grows significantly.

Interestingly, in adaptive settings, when sampling each $\boldsymbol{u}$, the attacker has to sample $m$ times with the same $\boldsymbol{u}$. Thus, the total number of samples an adaptive attacker needs is significantly higher; since $n' = n \times m$.

# B  PROOF FOR THEORETICAL ANALYSIS AGAINST GRADIENT-FREE ATTACKS

In this section, we provide the theoretical analysis of our defense method against gradient-free (search-based) attacks and the proof of proposition 2.

*Proof.* Given input $\boldsymbol{x}$, a constant $\epsilon > 0$, $\boldsymbol{u} \sim \mathcal{N}(0, \boldsymbol{I})$, the search direction of search-based attacks when considered against against the entire set of models (the ensemble or more generally a single model) relies on the sign of $\hat{H}(\boldsymbol{x}, \boldsymbol{u}) = F(\boldsymbol{x} + \epsilon\boldsymbol{u}) - F(\boldsymbol{x})$. Similarly, the search direction of search-based attacks against our defense employing different random subsets of models depends on the sign of $\tilde{H}(\boldsymbol{x}, \boldsymbol{u}) = q(\boldsymbol{x} + \epsilon\boldsymbol{u}; \boldsymbol{\pi}^{(i)}) - q(\boldsymbol{x}; \boldsymbol{\pi}^{(j)})$.

Applying Taylor expansion at $\boldsymbol{x}$, we have:

$$\hat{H}(\boldsymbol{x}, \boldsymbol{u}) = F(\boldsymbol{x} + \epsilon\boldsymbol{u}) - F(\boldsymbol{x}) \approx F(\boldsymbol{x}) + \epsilon\boldsymbol{u}\nabla F(\boldsymbol{x}) - F(\boldsymbol{x}) \approx \epsilon\boldsymbol{u}\nabla F(\boldsymbol{x})$$

Then, we can obtain:

$$\begin{aligned}
\tilde{H}(\boldsymbol{x}, \boldsymbol{u}) &= q(\boldsymbol{x} + \epsilon\boldsymbol{u}; \boldsymbol{\pi}^{(i)}) - q(\boldsymbol{x}; \boldsymbol{\pi}^{(j)}) \\
&\approx q(\boldsymbol{x}; \boldsymbol{\pi}^{(i)}) + \epsilon\boldsymbol{u}\nabla q(\boldsymbol{x}; \boldsymbol{\pi}^{(i)}) - q(\boldsymbol{x}; \boldsymbol{\pi}^{(j)}) \\
&\approx \left[ q(\boldsymbol{x}; \boldsymbol{\pi}^{(i)}) - q(\boldsymbol{x}; \boldsymbol{\pi}^{(j)}) \right] + \epsilon\boldsymbol{u}\nabla q(\boldsymbol{x}; \boldsymbol{\pi}^{(i)})
\end{aligned}$$

Thus:

$$\tilde{H}(\boldsymbol{x}, \boldsymbol{u}) - \hat{H}(\boldsymbol{x}, \boldsymbol{u}) \approx \underbrace{\left[ q(\boldsymbol{x}; \boldsymbol{\pi}^{(i)}) - q(\boldsymbol{x}; \boldsymbol{\pi}^{(j)}) \right]}_{\gamma_{i,j}} + \epsilon\boldsymbol{u} \underbrace{\left[ \nabla q(\boldsymbol{x}; \boldsymbol{\pi}^{(i)}) - \nabla F(\boldsymbol{x}) \right]}_{\zeta_i}$$

Following the proof of Theorem 3 in Qin et al. (2021), we can now obtain:

$$P(\frac{\tilde{H}(\boldsymbol{x}, \boldsymbol{u})}{\hat{H}(\boldsymbol{x}, \boldsymbol{u})} < 0) \leq P(|\tilde{H}(\boldsymbol{x}, \boldsymbol{u}) - \hat{H}(\boldsymbol{x}, \boldsymbol{u})| \geq |\hat{H}(\boldsymbol{x}, \boldsymbol{u})|)$$

$$\leq \frac{\mathbb{E}_{\boldsymbol{\pi}}\left[ |\tilde{H}(\boldsymbol{x}, \boldsymbol{u}) - \hat{H}(\boldsymbol{x}, \boldsymbol{u})| \right]}{|\hat{H}(\boldsymbol{x}, \boldsymbol{u})|} \quad \text{according to the Markov's inequality}$$

$$\leq \frac{\sqrt{\mathbb{E}_{\boldsymbol{\pi}}\left[ (\tilde{H}(\boldsymbol{x}, \boldsymbol{u}) - \hat{H}(\boldsymbol{x}, \boldsymbol{u}))^2 \right]}}{|\hat{H}(\boldsymbol{x}, \boldsymbol{u})|} \quad \text{according to the Jensen's inequality}$$

$$\leq \frac{\sqrt{\mathbb{E}_{\boldsymbol{\pi}}\left[ (\gamma_{i,j} + \epsilon\boldsymbol{u}\zeta_i)^2 \right]}}{|\hat{H}(\boldsymbol{x}, \boldsymbol{u})|}$$

$$\leq \frac{\sqrt{2\mathbb{E}_{\boldsymbol{\pi}}\left[ \gamma_{i,j}^2 + (\epsilon\boldsymbol{u}\zeta_i)^2 \right]}}{|\hat{H}(\boldsymbol{x}, \boldsymbol{u})|} \quad \text{according to the Cauchy's inequality}$$

$\blacksquare$

# C  ANALYSIS OF TRADE-OFF BETWEEN SUBSET SET SIZE AND ERROR ESTIMATION

In this section, we provide an additional analysis of the trade-off between the selection of $N$ (the subset set size) from $K$ models and the number of queries to achieve a low error estimation.

- Intuitively, a larger subset size $N$ reduces the *number* of combinations of model subsets. This results in a reduction in the number of random models presented to the attacker.

- In addition, a larger subset size $N$ also reduces the variance in estimates of gradient attempted by an attacker. Because, the prediction from a larger subset of models is more confident and the variance, for example, in output scores between these large subsets is less.

- Consequently, averaging across larger subsets of models leads to more informative responses (better gradient estimates, for example) and fewer queries to obtain low error estimations.

- In contrast, smaller $K$ values increase the uncertainty, which leads to increased variance in gradient estimation, or in other words, the difference in upper and lower bound for the gradient's value will be larger. Then following **Proposition 1**, this increases the cost of the attack, which forces the attacker to expend more queries to obtain a low error estimation of a gradient.

## D EFFECTIVENESS OF THE PROPOSED LEARNING OBJECTIVE

Table 7: Clean accuracy under different configurations (all models, random selection of subsets of individual models and each individual model). A comparison between a set of models trained simultaneously with and without the sample loss objective on `MNIST` (40 models), `CIFAR-10` (10 models) and `STL-10` (10 models).

| Dataset | MNIST | | CIFAR-10 | | STL-10 | |
|---|---|---|---|---|---|---|
| Training Objective | Without Sample Loss | With Sample Loss | Without Sample Loss | With Sample Loss | Without Sample Loss | With Sample Loss |
| All Models | **99.6%** | **99.7%** | **89.8%** | **93.2%** | **88.56%** | **89.93%** |
| Random Selection | | | | | | |
| 8 Models | 87.3% | **99.6%** | 59.7% | **92.8%** | 82.07% | **89.11%** |
| 5 Models | 79.4% | **99.6%** | 39.8% | **92.4%** | 78.41% | **88.46%** |
| 3 Models | 69.9% | **99.6%** | 31.0% | **91.4%** | 75.13% | **87.63%** |
| Individual Model or Parameter Particle Performance | | | | | | |
| Model 1 | 50.7% | **99.3%** | 15.1% | **88.5%** | 58.2% | **83.03%** |
| Model 2 | 36.6% | **99.5%** | 13.8% | **88.3%** | 58.69% | **81.44%** |
| Model 3 | 22.8% | **99.3%** | 13.5% | **88.5%** | 51.51% | **82.25%** |
| Model 4 | 42.2% | **99.2%** | 10.0% | **88.1%** | 51.7% | **84.79%** |
| Model 5 | 32.7% | **99.5%** | 9.3% | **88.9%** | 55.11% | **82.7%** |
| Model 6 | 35.4% | **99.4%** | 12.4% | **86.9%** | 60.39% | **84.88%** |
| Model 7 | 35.6% | **99.4%** | 11.3% | **88.4%** | 52.94% | **83.65%** |
| Model 8 | 32.0% | **99.4%** | 12.4% | **89.7%** | 43.85% | **83.79%** |
| Model 9 | 55.6% | **99.3%** | 10.2% | **87.7%** | 52.79% | **80.6%** |
| Model 10 | **99.3%** | **99.3%** | **80.8%** | **88.4%** | 71.08% | **82.78%** |

As we discussed in Section 3.4.2, incorporating sample loss as a training objective can encourage individual model learning and help each model obtain high performance because:

- Minimizing the loss over an average of logits for a subset of model faces the same problem we tried to address with the introduction of our learning objective (Sample Loss)—minimizing the loss over an average of logits for a subset of models promotes strong ensemble performance but does not guarantee that *each* individual model will perform well.

- Individual model performance, as we mentioned in Section 3.4.2, is very important to ensure minimal performance degradation for our defense. Because we want any randomly selected model or set to be well-performing.

- To this end, the proposed objective, through the joint training process, promotes diversity among models and ensures each individual model maintains strong performance.

The resulting *diverse* and *well-trained* models lead to the success of our proposed approach while minimizing impacts on clean accuracy. Therefore, in this section, we aim to show the effectiveness of and insights from the new training objective—sample loss—by considering models with and without sample loss.

We employ the SVGD method to train a set of models simultaneously, with and without sample loss for three tasks, `MNIST` (40 models), `CIFAR-10` (10 models) and `STL-10` (10 models). We train up to 1,000 epochs and select the best model set based on test accuracy. The results in Table 7 show that each individual model in a set trained with the sample loss objective achieves high performance on both datasets. As a result, any randomly selected five individual models are able to obtain high accuracy (92.4%) albeit slightly lower than the accuracy achieved by the set of *All Models* (93.2%). In contrast, without the sample loss objective, most models exceed 50% accuracy, and the random selection of five models does not result in high accuracy (79%).

## E    COST ANALYSIS

Our approach provides significant improvements in robustness. However, achieving robustness requires training (a one-time cost) and model storage cost. In this section, we analyze these costs and investigate a method for mitigating the cost. Followed by an experimental evaluation of the method with the high resolution `ImageNet` task.

Cost and complexity comparisons shown in Table 8 and Table 9 for training a single model versus sets of models as used in our experiments show that achieving better robustness does come at some cost.

Table 8: Training and inference times of different models between different defense mechanisms RND, RF and Disco(SVGD+) (40 models for `MNIST`, 10 models for `CIFAR-10/STL-10`). Here, for fairness, we assume the Disco methods process inputs one model at a time (*sequential*); in practice, the inference times can be similar to a single model as the forward pass of the input can occur in parallel across an ensemble.

| Datasets | Training Time | | Inference Time | | | |
|---|---|---|---|---|---|---|
| | Single Model (RND and RF) | A set of models (Disco) | Undefended | RND | RF | Disco (*sequential*) |
| MNIST | ∼0.5 hr | ∼12.5 hrs | 10.17 ms | 12.14 ms | 12.53 ms | 15.12 ms |
| CIFAR-10 | ∼1.5 hr | ∼72 hrs | 10.56 ms | 12.61 ms | 12.92 ms | 20.62 ms |
| STL-10 | ∼1.2 hr | ∼60 hrs | 11.26 ms | 13.12 ms | 13.48 ms | 24.85 ms |

Table 9: Trainable Parameters and Storage Consumption of models trained on different datasets between a single model and a set of models (Disco(SVGD+))—40 models for `MNIST`, 10 models for `CIFAR-10/STL-10`.

| Datasets | Trainable Parameters | | Storage Consumption | |
|---|---|---|---|---|
| | Single Model (RND and RF) | A set of models (Disco) | Single Model (RND and RF) | A set of models (Disco) |
| MNIST | 0.312 M | 12.5 M | 1.19 MB | 47.7 MB |
| CIFAR-10 | 14.73 M | 147.3 M | 56.18 MB | 561.84 MB |
| STL-10 | 11.18 M | 111.8 M | 43.12 MB | 426.55 MB |

## F    COST MITIGATION METHOD

In general, the use of multiple models does lead to increasing the training and storage burden. RND and RF use a single model, whereas we employ a set of $n$ models, so the number of parameters in our approach is $n\times$ higher, and the memory consumption is also larger. In practical applications, the cost can be mitigated:

- Recent work research in the area of model tuning with low-rank adapters (LoRAs) Hu et al. (2021) can mitigate the cost of building large-scale practical ensembles.
- The study in Doan et al. (2024) develops a method for a pre-trained model to be tuned with only a $1\%$ increase in trainable parameters and storage costs to build ensembles of diverse

models using SVGD. The authors employ the pre-trained OpenCLIP Radford et al. (2021) model for the `ImageNet` task and LlaVA for a visual question and answer task.

Next, we adopt the method of model fine tuning to demonstrate how `Disco()`, the model randomization method, can be implemented for *practical tasks*, represented by `ImageNet`, and *for a practical, large-scale model*, represented by OpenCLIP.

### F.1 EFFECTIVENESS ON HIGH-RESOLUTION LARGE-SCALE DATASET AND THE PRACTICAL LARGE-SCALE **OPENCLIP** MODEL

In this section, we demonstrate the effectiveness of our method on high-resolution datasets like `ImageNet` and with a large-scale, piratical model, *OpenCLIP* Radford et al. (2021).

**Robustness Comparison**

Inspired by recent work Doan et al. (2024), we adopt the technique of fine-tuning pre-trained models to obtain well-trained, large-scale models at a fraction of the cost of training an ensemble from initialization. The authors employ SVGD to achieve model diversity and use low-rank adapters to significantly reduce the cost of building the ensemble to better approximate a multi-modal Bayesian posterior.

As a demonstration of a practical application with a large-scale model, we also use the pre-trained, OpenCLIP, large-scale model with low-rank adaptors (LoRA) Hu et al. (2021) as in Doan et al. (2024) to build a sample of five models for the `ImageNet` task. The ensemble achieves approximately 78% clean accuracy on the test set. We used a random selection of two out of five models in our method for the defense, where the clean accuracy of two out of 5 models is approximately 77%. For RND and RF defenses, we fine-tune the CLIP model for the `ImageNet` task to achieve 76.07 % clean accuracy and, for a fair comparison, we choose hyperparameters such that the clean accuracy drop, after injecting noise, is approximately 1%. In this experiment, we randomly select 100 correctly classified images from the `ImageNet` test set and use the SQUAREATTACK ($l_\infty$) against the models.

The results in Table 10 show that our approach achieves the best results across various perturbation budgets on `ImageNet` task compared to both RND and RF methods. Importantly, our approach is able to achieve up to 9.6% increase in robustness above the next best performing method RF with an ensemble of just 5 models.

Table 10: $l_\infty$ **objective**. Robustness (higher ↑ is stronger) of defenses against SQUAREATTACK on the `ImageNet` task with an OpenCLIP model.

| Methods | $l_\infty$ =0.025 | 0.05 | 0.075 | 0.1 |
|---|---|---|---|---|
| RND | 83.39% | 61.95% | 43.37% | 24.89% |
| RF | 86.45% | 65.1 % | 51.14% | 35.83% |
| **Disco** | **90.76%** | **72.51%** | **56.17%** | **45.4%** |

**Cost Mitigation Analysis**

The results in Table 11 show that our approach can be realized in large-scale network architecture like OpenCLIP and yet achieve the best robustness results across various perturbation budgets on the `ImageNet` task compare to both RND and RF methods. Now, only a marginal cost increase is needed to achieve significant improvements in robustness.

Table 11: Trainable Parameters and Storage Consumption of five OpenCLIP with LoRA trained on `ImageNet`. Notably, we begin with a single *pre-trained* OpenCLIP model and subsequently construct the ensemble of five models while tuning the model for the `ImageNet` task.

| Models | Single CLIP | A set of CLIPs with LoRA |
|---|---|---|
| Trainable Parameters | 114 M | 1.84 M (1.6%, 0.32% per model) |
| Storage Consumption | 433 MB | 439 MB (↑1.38%, 0.28% per model) |

**Summary**

What we are proposing are *marginal* cost increases to achieve significant improvements in robustness.

- Effectively, we demonstrate $<1.6\%$ increase in overhead can yield up to $9.6\%$ better robustness (when compared to the next best defense method) on a large-scale network of practical significance.
- Now, adding a model incurs $<0.32\%$ overhead in terms of trainable parameters or storage.

Overall, these results also demonstrate that our model randomization method is:

- Practical for implementation
- Effective across different datasets and model types.

## G  EFFECTIVENESS AGAINST AN ATTACK IN A SURROGATE MODEL SETTING

In this section, we further assess the robustness of the different defense mechanisms against the state-of-the-art attack using a surrogate model. In the Prior-Bayesian Optimization (P-BO) attack Cheng et al. (2024), it integrates transfer-based and query-based techniques. The results in Table 12 demonstrate that our defense outperforms RND and RF and effectively fortifies against the strong P-BO attack setting. This underpins the capability of Disco to withstand even the most advanced query-based attacks. This reinforces the strength and general applicability of our approach in defending against cutting-edge black-box attack methods such as P-BO.

Table 12: $l_\infty$ **objective**. Robustness (higher $\uparrow$ is stronger) of defenses against P-BO with `CIFAR-10` task.

| Methods | $l_\infty =0.02$ | 0.04 | 0.06 | 0.08 | 0.1 |
|---------|---------|------|------|------|-----|
| Single (*undef*) | 0.0% | 0.0% | 0.0% | 0.0% | 0.0% |
| RND | 70.33% | 31.47% | 15.75% | 7.23% | 6.65% |
| RF | 66.43% | 28.04 % | 13.67% | 8.34% | 6.21% |
| **Disco** | **79.98**% | **47.94**% | **29.8**% | **18.08**% | **12.16**% |

## H  ALTERNATIVE APPROACHES FOR MODEL DIVERSITY PROMOTION

### H.1  FORMULATION OF TRAINING OBJECTIVES

**Ensembles employing Random Initialization Approaches.** Lakshminarayanan et al. (2017) proposed to train a set of models—*Ensemble*—with random initializations independently. This training is formulated as follows:

$$\min_{\boldsymbol{\theta}_k} \quad \mathbb{E}_{(\boldsymbol{x},y)\sim\mathcal{D}}\Big[\ell(f_k(\boldsymbol{x};\boldsymbol{\theta}_k),y)\Big], \tag{13}$$

where $\boldsymbol{\theta}_i$ denotes the weights of the $i$-th model, and $\ell(.,.)$ is the loss (*i.e.* cross-entropy).

**Gradient-based Approach.** Teney et al. (2022) introduced a method encouraging diversity over a set of models by quantifying the similarity of the gradient of the top predicted score of each model with respect to its features. This method aims to train a model set to discover predictive patterns commonly missed by learning algorithms and promote diversity across the model set. In our study, we adopt their *Diversity Regularizer* (DivReg) to encourage the model diversity and formulate the training objective as follows:

$$\min_{\Theta} \mathbb{E}_{(\boldsymbol{x},y)\sim\mathcal{D}}\Big[ \sum_{k=1}^{K} \ell(f_k(\boldsymbol{x};\boldsymbol{\theta}_k),y) + \lambda_{\text{reg}} \sum_{i\neq j} \delta_{f_i,f_j} \Big], \tag{14}$$

where $\delta_{f_i,f_j} = \nabla_h f_i(\boldsymbol{h}_i) \cdot \nabla_h f_j(\boldsymbol{h}_j)$, $\lambda_{\text{reg}}$ controls the strength of the regularizer, $\nabla_h f_i$ and $\nabla_h f_j$ denote the gradient of the top predicted score of models $f_i$ and $f_j$ w.r.t their own features $\boldsymbol{h}_i$ and $\boldsymbol{h}_j$.

**Score-based Approach.** Lee et al. (2023) proposed an approach to training a collection of diverse models by independently training each head pair to make predictions. In our study, we adopt their loss function to encourage model diversity. The training objective is formulated as follows:

$$\min_{\boldsymbol{\Theta}} \mathbb{E}_{(\boldsymbol{x},y)\sim\mathcal{D}} \Big[ \sum_{k=1}^{K} \ell(f_k(\boldsymbol{x};\boldsymbol{\theta}_k), y) + \lambda_{\mathrm{MI}} \sum_{k\neq i} \mathcal{L}_{\mathrm{MI}}(f_k(\boldsymbol{x};\boldsymbol{\theta}_k), f_i(\boldsymbol{x};\boldsymbol{\theta}_i)) \Big], \qquad (15)$$

where $\mathcal{L}_{\mathrm{MI}}(f(\boldsymbol{x};\boldsymbol{\theta}_k), f(\boldsymbol{x};\boldsymbol{\theta}_i)) = D_{KL}(p(\hat{y_k}, \hat{y_i}) \parallel p(\hat{y_k}) \otimes p(\hat{y_i}))$, $D_{\mathrm{KL}}(. \parallel .)$ is the KL divergence and $\hat{y_i}$ is the predicted label from $f_i(\boldsymbol{x};\boldsymbol{\theta}_i)$, $\lambda_{\mathrm{MI}}$ controls the strength of mutual information loss $\mathcal{L}_{\mathrm{MI}}$, $p(\hat{y_k}, \hat{y_i})$ is the empirical estimate of the joint distribution and $p(\hat{y_k})$, $p(\hat{y_i})$ are the empirical estimates of the marginal distributions.

## H.2 DIVERSITY ANALYSIS

As we discussed in Section 3.4, more diversity among individual models or particles results in enhanced output diversity. Therefore, we use Jensen–Shannon divergence as a metric to illustrate model diversity promoted by different methods. We measure diversity by calculating the Jensen–Shannon divergence between the average softmax outputs of all models $\hat{p}(x_i) = \frac{1}{K}\sum_{k=1}^{K} \mathrm{softmax}[f(\boldsymbol{x}_i;\boldsymbol{\theta}_k)]$ versus the softmax output of each individual model $p_k(x_i) = \mathrm{softmax}[f(\boldsymbol{x_i};\boldsymbol{\theta}_k)]$ and then averaging across all samples from a test set as follows:

$$D_{JS}^{(k)} = \frac{1}{2} \sum_{i}^{N} \Big( D_{KL}\Big(\hat{p}(x_i) \parallel \frac{\hat{p}(x_i) + p_k(x_i)}{2}\Big) + D_{KL}\Big(p_k(x_i) \parallel \frac{\hat{p}(x_i) + p_k(x_i)}{2}\Big) \Big),$$

where $D_{\mathrm{KL}}(. \parallel .)$ is the KL divergence, k represent the individual model $k$-th and $N$ denotes the size of a dataset. The results as shown in Section 4.4 and Figure 3 demonstrate that our proposed method (SVGD+) is able to achieve greater diversity among individual parameter particles. *These empirical findings support the assertions of our hypotheses in Section 3 and the robustness of the defense method we formulated.*

## H.3 DIVERSITY ANALYSIS OF INDIVIDUAL MODELS LEARNED WITH DIFFERENT TRAINING OBJECTIVES

Similar to Section 4.1, we measure the diversity between every pair of individual models which can be computed with Equation 16. The results demonstrated in Figure 4 show highly diverse nature of the models (larger range of colors for model vs. model results) and that each individual model trained by our proposed approach obtains higher diversity (denoted by lighter colors).

$$D_{JS}^{(k,j)} = \frac{1}{2} \sum_{i}^{N} \Big( D_{KL}\Big(p_k(x_i) \parallel \frac{p_k(x_i) + p_j(x_i)}{2}\Big) + D_{KL}\Big(p_j(x_i) \parallel \frac{p_k(x_i) + p_j(x_i)}{2}\Big) \Big), \quad (16)$$

Table 13: **CIFAR−10**. Robustness (higher ↑ is stronger) of different defense methods against NE-SATTACK, SIGNHUNTER and SQUAREATTACK attacks under the $l_\infty$ perturbation objective.

| Attack | Methods | $l_\infty$ =0.02 | 0.04 | 0.06 | 0.08 | 0.1 |
|---|---|---|---|---|---|---|
| NESATTACK | Disco(Ensemble) | 98.82% | 95.32 % | 89.77% | 85.43% | 81.11% |
| | Disco(DivDis) | 99.51% | **98.5** % | **95.05**% | **92.64**% | **88.54**% |
| | Disco(DivReg) | 99.45% | 97.54 % | 92.94% | 89.53% | 84.65% |
| | **Disco(SVGD+)** | 99.7% | 97.93% | 94.39% | 90.5% | 86.77% |
| SIGNHUNTER | Disco(Ensemble) | 99.88% | 96.95 % | 90.03% | 81.75% | 73.79% |
| | Disco(DivDis) | 99.88% | 98.11 % | 92.83% | 84.97% | 77.44% |
| | Disco(DivReg) | 99.98% | 98.30 % | 90.13% | 77.4% | 64.86% |
| | **Disco(SVGD+)** | 99.97% | **98.95**% | **95.56**% | **90.7**% | **84.22**% |
| SQUAREATTACK | Disco(Ensemble) | 99.77% | 91.08 % | 71.69% | 52.67% | 39.69% |
| | Disco(DivDis) | 99.96% | 96.85 % | 85.23% | 67.96% | 52.29% |
| | Disco(DivReg) | **99.99**% | 96.06 % | 83.1% | 64.1% | 48.63% |
| | **Disco(SVGD+)** | 99.97% | **96.91**% | **86.52**% | **70.22**% | **55.77**% |

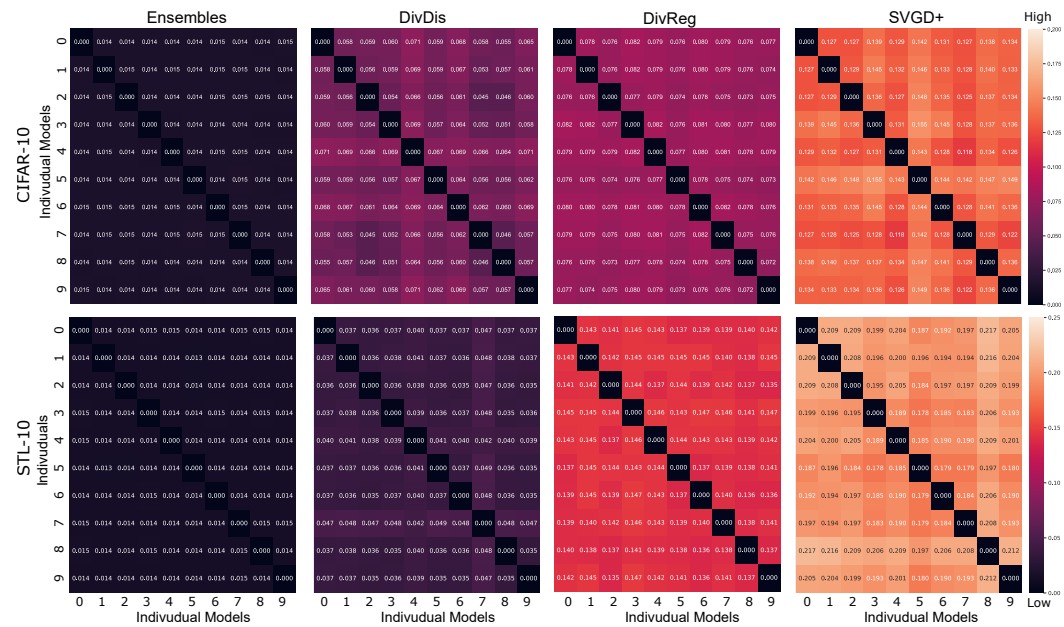

Figure 4: The diversity measurement (using Jensen–Shannon divergence) between every pair of individual models trained by Ensemble, DivDis, DivReg and our proposed method SVGD+ (Ours) on `CIFAR-10` and `STL-10`.

# I ROBUSTNESS EVALUATION OF ALTERNATIVE APPROACHES

In this section, we conduct extensive experiments to evaluate the robustness of alternative approaches for model diversity promotion (Ensemble, DivDis and DivReg) together with our proposal (SVGD+) against score-based adversarial attack NESATTACK ($l_\infty$), SIGNHUNTER ($l_\infty$), SQUAREATTACK ($l_\infty$), SPARSERS ($l_0$) and decision-based attacks HOPSKIPJUMP ($l_2$) and SPAEVO ($l_0$) on `CIFAR-10`. The results in Tables 13, 14 and 15 demonstrate that our proposed defense mechanism outperforms all other alternative approaches.

Table 14: **`CIFAR-10`**. Robustness (higher ↑ is stronger) of different model diversity promotion schemes against SPARSERS ($l_0$).

| Methods | $l_0$ =16px | 32px | 48px | 64px | 80px |
|---|---|---|---|---|---|
| Disco(Ensemble) | 50.12% | 36.82% | 25.97% | 23.16% | 19.06% |
| Disco(DivDis) | 57.94% | 41.2% | 35.15% | 29.75% | 25.59% |
| Disco(DivReg) | 61.38% | 44.95% | 39.17% | 32.62% | 28.0% |
| **Disco(SVGD+)** | **63.85**% | **47.84**% | **41.59**% | **36.81**% | **31.24**% |

Table 15: **Decision-based, `CIFAR-10`**. Robustness (higher ↑ is stronger) of different model diversity promotion schemes against HOPSKIPJUMP ($l_2$) and SPAEVO ($l_0$).

| Methods | HOPSKIPJUMP | | | | | SPAEVO | | | | |
|---|---|---|---|---|---|---|---|---|---|---|
| | $l_2$ =0.8 | 1.6 | 2.4 | 3.2 | 4.0 | $l_0$ =4px | 8px | 12px | 16px | 20px |
| Disco(Ensemble) | 99.82% | 99.12 % | 97.94% | 96.95% | 96.31% | 92.07% | 91.79 % | 91.69% | 91.67% | 91.66% |
| Disco(DivDis) | 99.94% | **99.59** % | **98.94**% | 98.04% | 97.33% | 93.68% | 93.3% | 93.1% | 93.07% | 92.97% |
| Disco(DivReg) | 99.98% | 99.25 % | 98.26% | 97.39% | 96.41% | 93.9% | 93.53% | 93.42% | 93.35% | 93.32% |
| **Disco(SVGD+)** | **99.94**% | 99.4% | 98.77% | **98.04**% | **97.43**% | **96.17**% | **95.99**% | **95.94**% | **95.88**% | **95.84**% |

Table 16: **MNIST**. A robustness comparison (higher ↑ is stronger) between our proposed method and other methods against SQUAREATTACK. For the evaluation of different diversity-promotion methods, we train a set of 10 models and randomly select a subset of a different number of models.

| Random | Methods | $l_2$=0.8 | 1.6 | 2.4 | 3.2 | 4.0 |
|---|---|---|---|---|---|---|
| 1 | Disco(Ensemble) | 98.4% | 91.9% | 87.1% | 82.0% | 74.6% |
| | Disco(DivDis) | 98.3% | 92.9% | 86.6% | 78.9% | 70.9% |
| | Disco(DivReg) | 96.1% | 88.5% | 80.9% | 72.5% | 66.5% |
| | **Disco(SVGD+)** | **98.7%** | **94.4%** | **89.3%** | **86.0%** | **80.0%** |
| 3 | Disco(Ensemble) | 99.9% | 98.7% | 91.6% | 78.7% | 68.0% |
| | Disco(DivDis) | 99.8% | 97.6% | 87.8% | 76.9% | 62.5% |
| | Disco(DivReg) | 99.8% | 85.5% | 88.1% | 77.6% | 69.1% |
| | **Disco(SVGD+)** | **99.9%** | **99.6%** | **95.4%** | **83.3%** | **70.3%** |
| 5 | Disco(Ensemble) | 100% | 98.6% | 90.5% | 74.4% | 55.8% |
| | Disco(DivDis) | 99.8% | 96.3% | 85.0% | 71.6% | 57.0% |
| | Disco(DivReg) | 99.9% | 94.4% | 83.8% | 72.1% | 60.0% |
| | **Disco(SVGD+)** | **100%** | **99.4%** | **95.3%** | **81.1%** | **63.8%** |
| 8 | Disco(Ensemble) | 100% | 98.5% | 90.3% | 73.5% | 54.2% |
| | Disco(DivDis) | 99.5% | 94.3% | 82.6% | 67.1% | 53.5% |
| | Disco(DivReg) | 99.9% | 93.2% | 80.7% | 72.3% | 59.7% |
| | **Disco(SVGD+)** | **100%** | **99.4%** | **95.3%** | **81.1%** | **63.8%** |

# J EVALUATIONS WITH DIFFERENT RANDOMIZED MODEL SELECTION STRATEGIES

## J.1 ON MNIST

In this section, we provide additional results for training a set of 10, 20 and 40 models using Ensemble, DivDis, DivReg and our proposed method under SQUAREATTACK ($l_2$). Table 20 shows clean accuracy under different model training and random selection strategies. For robustness evaluation and comparison, we choose different settings with different sizes of model subsets. For instance, we sample 1, 3, 5, 8 of 10 models and sample 1, 3, 5, 10 of 20 models. For 40 models, we sample 1, 3, 5, 20 and 30 of 40 models. The results in Tables 16, 17 and 18 provide further evidence to demonstrate that our proposed method is more robust than other diversity promotion methods across different distortions and settings.

Table 17: **MNIST**. A robustness comparison (higher ↑ is stronger) between our proposed method and other methods against SQUAREATTACK. For the evaluation of different diversity-promotion methods, we train a set of 20 models and randomly select a subset of a different number of models.

| Random | Methods | $l_2$= 0.8 | 1.6 | 2.4 | 3.2 | 4.0 |
|---|---|---|---|---|---|---|
| 1 | Disco(Ensemble) | 99.2% | 96.1% | 91.6% | 85.1% | 75.4% |
| | Disco(DivDis) | 99.0% | 95.5% | 91.1% | 82.7% | 74.5% |
| | Disco(DivReg) | 96.5% | 91.8% | 83.7% | 76.8% | 68.6% |
| | **Disco(SVGD+)** | **99.4%** | **97.3%** | **94.2%** | **90.2%** | **83.5%** |
| 3 | Disco(Ensemble) | 100% | 99.5% | 95.4% | 85.7% | 70.8% |
| | Disco(DivDis) | 100% | 97.8% | 89.9% | 79.3% | 66.5% |
| | Disco(DivReg) | 100% | 97.0% | 91.1% | 83.9% | 76.5% |
| | **Disco(SVGD+)** | **100%** | **99.8%** | **98.1%** | **90.5%** | **78.8%** |
| 5 | Disco(Ensemble) | 100% | 99.3% | 93.2% | 77.5% | 62.8% |
| | Disco(DivDis) | 99.8% | 97.5% | 90.6% | 76.8% | 60.4% |
| | Disco(DivReg) | 99.9% | 97.8% | 90.6% | 76.8% | 60.4% |
| | **Disco(SVGD+)** | **100%** | **99.4%** | **94.8%** | **85.1%** | **70.0%** |
| 10 | Disco(Ensemble) | 99.9% | 98.2% | 86.5% | 67.1% | 46.0% |
| | Disco(DivDis) | 99.7% | 93.1% | 79.1% | 65.0% | 50.1% |
| | Disco(DivReg) | 99.9% | 94.0% | 83.0% | 72.9% | **62.6%** |
| | **Disco(SVGD+)** | **100%** | **99.5%** | **95.0%** | **76.8%** | 60.0% |

## J.2 ON CIFAR-10

In this section, we provide additional results for robustness evaluation and comparison in different settings with different sizes of model subsets (*i.e.* 1, 3, 5, and 8) under SQUAREATTACK ($l_2$). The results in Table 19 show that our proposed method is more robust than other diversity promotion methods across different distortions and settings.

Table 18: **MNIST**. A comparison of robustness (higher ↑ is better) between Disco(SVGD+) and other learning methods against SQUAREATTACK. For the evaluation of different diversity promotion methods, we train a set of 40 models and randomly select a subset of a different number of models.

| Random | Methods | $l_2 = 0.8$ | 1.6 | 2.4 | 3.2 | 4.0 |
|---|---|---|---|---|---|---|
| 1 | Disco(Ensemble) | 99.6% | 97.3% | 93.4% | 89.0% | 80.2% |
| | Disco(DivDis) | 99.6% | 97.4% | 93.9% | 88.1% | 82.6% |
| | Disco(DivReg) | 99.2% | 96.2% | 91.8% | 84.3% | 77.4% |
| | **Disco(SVGD+)** | **99.7%** | **98.9%** | **97.2%** | **93.5%** | **88.2%** |
| 3 | Disco(Ensemble) | 100% | 99.4% | 94.2% | 85.2% | 74.6% |
| | Disco(DivDis) | 100% | 98.6% | 93.8% | 83.7% | 73.1% |
| | Disco(DivReg) | 100% | 99.0% | 93.0% | 79.7% | 67.6% |
| | **Disco(SVGD+)** | **100%** | **99.8%** | **98.0%** | **91.4%** | **77.8%** |
| 5 | Disco(Ensemble) | 100% | 99.5% | 95.8% | 84.9% | 70.8% |
| | Disco(DivDis) | 100% | 98.6% | 94.3% | 79.9% | 68.9% |
| | Disco(DivReg) | 100% | 98.4% | 90.5% | 79.5% | 67.9% |
| | **Disco(SVGD+)** | **100%** | **99.8%** | **97.9%** | **90.1%** | **76.5%** |
| 20 | Disco(Ensemble) | 100% | 97.6% | 86.4% | 68.0% | 49.0% |
| | Disco(DivDis) | 99.8% | 95.9% | 85.5% | 72.2% | 54.8% |
| | Disco(DivReg) | 99.7% | 96.1% | 83.3% | 67.0% | 51.3% |
| | **Disco(SVGD+)** | **100%** | **99.3%** | **94.4%** | **77.5%** | **56.8%** |
| 30 | Disco(Ensemble) | 99.9% | 96.8% | 81.2% | 60.6% | 40.0% |
| | Disco(DivDis) | 99.9% | 95.9% | 80.0% | 64.9% | 46.9% |
| | Disco(DivReg) | 99.5% | 93.9% | 77.3% | 59.7% | 43.2% |
| | **Disco(SVGD+)** | **100%** | **98.6%** | **91.9%** | **70.4%** | **52.2%** |

Table 19: **CIFAR-10**. A robustness comparison (higher ↑ is stronger) between our approach and other methods against SQUAREATTACK. For the evaluation of different diversity promotion methods, we train a set of 10 models and randomly select a subset of a different number of models.

| Random | Methods | $l_2 = 0.8$ | 1.6 | 2.4 | 3.2 | 4.0 |
|---|---|---|---|---|---|---|
| 1 | Disco(Ensemble) | 90.0% | 83.6% | 75.4% | 64.2% | 55.1% |
| | Disco(DivDis) | **95.1%** | **90.1%** | **82.6%** | 72.0% | 59.1% |
| | Disco(DivReg) | 90.6% | 86.2% | 79.5% | 69.6% | 59.5% |
| | **Disco(SVGD+)** | 90.2% | 86.9% | 82.2% | **75.2%** | **67.6%** |
| 3 | Disco(Ensemble) | 97.1% | 88.3% | 78.6% | 67.4% | 55.4% |
| | Disco(DivDis) | 99.2% | 96.1% | 86.2% | 75.8.3% | 62.1% |
| | Disco(DivReg) | 99.6% | 93.5% | 84.3% | 72.1% | 60.3% |
| | **Disco(SVGD+)** | **99.8%** | **96.7%** | **90.0%** | **82.2%** | **72.6%** |
| 5 | Disco(Ensemble) | 97.7% | 89.0% | 76.1% | 63.1% | 52.3% |
| | Disco(DivDis) | 99.0% | 93.9% | 83.0% | 70.8% | 55.7% |
| | Disco(DivReg) | 99.0% | 91.6% | 78.5% | 65.3% | 53.6% |
| | **Disco(SVGD+)** | **99.6%** | **95.6%** | **87.1%** | **76.5%** | **65.8%** |
| 8 | Disco(Ensemble) | 98.2% | 87.9% | 76.9% | 63.5% | 52.2% |
| | Disco(DivDis) | 99.1% | 94.1% | 82.7% | 70.4% | 56.4% |
| | Disco(DivReg) | 99.2% | 90.9% | 76.3% | 64.6% | 51.6% |
| | **Disco(SVGD+)** | **99.7%** | **96.0%** | **86.2%** | **76.2%** | **66.0%** |

### J.3 CLEAN ACCURACY OF DIFFERENT SUBSET

We demonstrate clean accuracy obtained by models trained by different model diversity promotion methods with different selection configurations in Table 20.

Table 20: Clean accuracy achieved by different defended models employing diversity-promotion techniques on different datasets with a different random number of models.

| MNIST | | | | | |
|---|---|---|---|---|---|
| Quantity | Random | Disco(Ensemble) | Disco(DivDis) | Disco(DivReg) | Disco(SVGD+) |
| 10 | 1 | 99.5% | 99.5% | 97.5% | 99.4% |
| | 3 | 99.6% | 99.6% | 99.2% | 99.6% |
| | 5 | 99.7% | 99.6% | 99.4% | 99.6% |
| | 8 | 99.6% | 99.6% | 99.5% | 99.6% |
| 20 | 1 | 99.5% | 99.5% | 97.6% | 99.0% |
| | 3 | 99.6% | 99.6% | 99.3% | 99.5% |
| | 5 | 99.6% | 99.6% | 99.4% | 99.5% |
| | 10 | 99.7% | 99.6% | 99.6% | 99.6% |
| 40 | 1 | 99.3% | 99.5% | 98.6% | 98.7% |
| | 3 | 99.5% | 99.5% | 99.4% | 99.2% |
| | 5 | 99.5% | 99.6% | 99.5% | 99.4% |
| | 20 | 99.6% | 99.7% | 99.6% | 99.6% |
| | 30 | 99.6% | 99.7% | 99.6% | 99.6% |
| CIFAR-10 | | | | | |
| Quantity | Random | Disco(Ensemble) | Disco(DivDis) | Disco(DivReg) | Disco(SVGD+) |
| 10 | 1 | 92.2% | 90.5% | 91.8% | 87.9% |
| | 3 | 93.8% | 92.5% | 93.9% | 91.1% |
| | 5 | 94.0% | 93.3% | 94.3% | 92.3% |
| | 8 | 94.4% | 93.5% | 94.5% | 92.5% |
| STL-10 | | | | | |
| Quantity | Random | Disco(Ensemble) | Disco(DivDis) | Disco(DivReg) | Disco(SVGD+) |
| 10 | 5 | 91.6% | 90.2% | 89.7% | 88.2% |

## K ROBUSTNESS OVER MULTIPLE TRIALS

In this section, we conduct an extensive experiment to study the robustness of different defense mechanisms with randomness involvement. We evaluate RND, RF, and our proposed method against SQUAREATTACK ($l_2$) on an evaluation set of 500 correctly classified images drawn from CIFAR-10. Each defense is evaluated five times with different random seeds. Figure 5 presents the mean accuracy under attacks, with the upper and lower error bars representing the mean $\pm$ standard deviation. The results in Figure 5 show that the variation of our method is similar to other defenses and our lower error bar is far higher than the upper error bar of both RND and RF.

## L COMPARISON WITH ADVERSARIAL TRAINING (AT)

We conduct an experiment to demonstrate the robustness of our proposed method Disco, RND, RF and a state-of-the-art adversarial training (AT) (Wang et al., 2023) used for the CIFAR-10 task. We used the strong, query-based black-box attack, SQUAREATTACK under the $l_2$ objective. We use the implementation and the pre-trained model ($l_2$) from *Robustbench*[2] (Croce et al., 2021).

The results in Figure 6, demonstrate that our simpler approach employing model radomization is better than the state-of-the-art adversarial training for a query-based black-box attack. Notably, our result comparison with AT also confirms those found in the recent black-box defense, RF Nguyen et al. (2024) where the AT methods itself was not as robust at the dedicated black-box defense (see AT vs. Ours in Table 4 in Nguyen et al. (2024)).

---

[2]https://github.com/RobustBench/robustbench

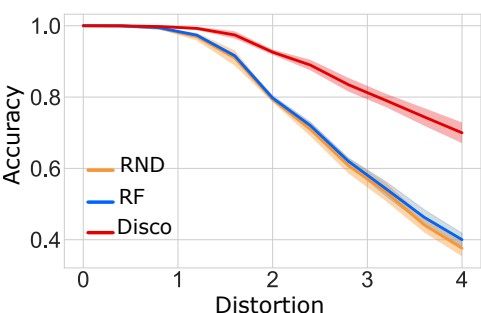

Figure 5: A comparison of the average robustness between different defenses against SQUAREAT-TACK ($l_2$). Mean accuracy under attacks, with the upper and lower error bars representing the mean $\pm 1\sigma$ (standard deviation).

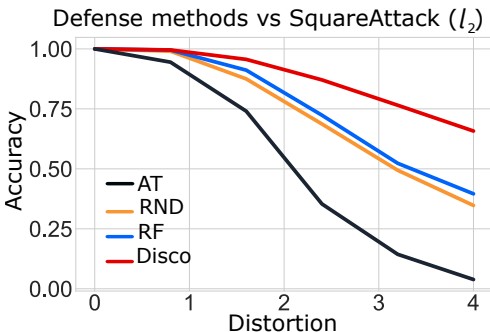

Figure 6: A comparison between Disco, RND, RF and AT against SQUAREATTACK ($l_2$).

Notably, it should be mentioned here that the model randomization methods can also be adopted with adversarial trained models. We expect the robustness afforded by AT methods to then further improve the robustness from our approach. However, employing AT methods are likely to come at the cost of noticeable clean accuracy drops.

## M    COMPARISON WITH ADVERSARIAL ATTACK ON ATTACKERS (AAA)

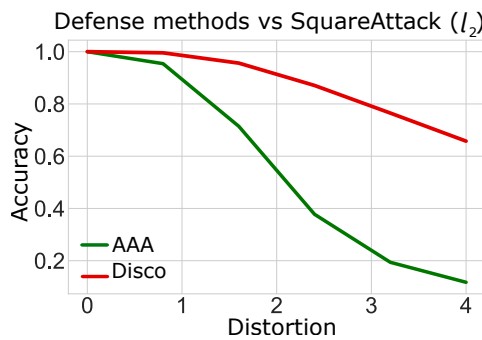

Figure 7: A robustness comparison between AAA and our method against SQUAREATTACK ($l_2$).

AAA is a defense algorithm mainly designed for *score-based* attacks so it is expected to be successful against these attacks. Notably, it does not strongly withstand decision-based attacks as reported in evaluations by Nguyen et al. (2024). Nevertheless, we conduct an experiment to demonstrate the robustness of our proposed method versus AAA (Chen et al., 2022) on `CIFAR-10` against the strong query-based black-box SQUAREATTACK ($l_2$) under the score-based setting. The results in Figure 7 demonstrate that our proposed defense mechanism is much more robust than AAA, especially at high perturbation budgets.

# N COMPARISON WITH REGION-BASED CLASSIFICATION (RBC)

Region-Based Classification (RBC) method (Cao & Gong, 2017) is a defense initially designed and evaluated with white-box attacks. It aims to add noise to the input and employ majority voting to make decisions. Given its strategy of adding random-noise to inputs, it can also be employed with black-box attacks. Therefore, in this section, we conduct extensive experiments to compare the robustness of our defense and the RBC method against score-based adversarial attacks NESATTACK ($l_\infty$), SIGNHUNTER ($l_2, l_\infty$), SQUAREATTACK ($l_2, l_\infty$), SPARSERS ($l_0$) and decision-based attacks HOPSKIPJUMP ($l_2$) and SPAEVO ($l_0$) on CIFAR-10. The results in tables 21, 22, 23 and 24 show that our method is more robust than both RBC.

Table 21: $l_2$ **objective attacks, CIFAR-10.** Robustness (higher ↑ is stronger) of different defense methods against SIGNHUNTER and SQUAREATTACK.

| Methods | SIGNHUNTER | | | | | SQUAREATTACK | | | | |
|---|---|---|---|---|---|---|---|---|---|---|
| | $l_2$ =0.8 | 1.6 | 2.4 | 3.2 | 4.0 | $l_2$ =0.8 | 1.6 | 2.4 | 3.2 | 4.0 |
| RBC | 25.86% | 10.34% | 3.55% | 1.79% | 1.18% | 8.64% | 2.65 % | 0.95% | 0.28% | 0.05% |
| **Disco** | **99.96**% | **99.25**% | **97.61**% | **93.63**% | **90.24**% | **99.56**% | **95.62**% | **87.07**% | **76.5**% | **65.76**% |

Table 22: $l_\infty$ **objective attacks, CIFAR-10.** Robustness (higher ↑ is stronger) of different defense methods against attacks NESATTACK, SIGNHUNTER and SQUAREATTACK.

| Attack | Methods | $l_\infty$ =0.02 | 0.04 | 0.06 | 0.08 | 0.1 |
|---|---|---|---|---|---|---|
| NESATTACK | RBC | 94.06% | 86.07 % | 79.08% | 76.03% | 73.34% |
| | **Disco** | **99.7**% | **97.93**% | **94.39**% | **90.5**% | **86.77**% |
| SIGNHUNTER | RBC | 29.69% | 6.99 % | 3.75% | 0.93% | 0.53% |
| | **Disco** | **99.97**% | **98.95**% | **95.56**% | **90.7**% | **84.22**% |
| SQUAREATTACK | RBC | 37.4% | 6.32 % | 2.63% | 0.1% | 0.1% |
| | **Disco** | **99.97**% | **96.91**% | **86.52**% | **70.22**% | **55.77**% |

Table 23: $l_0$ **objective attacks, CIFAR-10.** Robustness (higher ↑ is stronger) of different model diversity promotion schemes against SPARSERS.

| Methods | $l_0$ =16px | 32px | 48px | 64px | 80px |
|---|---|---|---|---|---|
| RBC | 0.43% | 0.29% | 0.2% | 0.17% | 0.01% |
| **Disco** | **63.85**% | **47.84**% | **41.59**% | **36.81**% | **31.24**% |

Table 24: **Decision-based.** Robustness (higher ↑ is stronger) of different model diversity promotion schemes against HOPSKIPJUMP ($l_2$) and SPAEVO ($l_0$) on the CIFAR-10 task.

| Methods | HOPSKIPJUMP | | | | | SPAEVO | | | | |
|---|---|---|---|---|---|---|---|---|---|---|
| | $l_2$ =0.8 | 1.6 | 2.4 | 3.2 | 4.0 | $l_0$ =4px | 8px | 12px | 16px | 20px |
| RBC | 94.44% | 79.24 % | 61.47% | 57.54% | 57.89% | 55.21% | 31.18% | 16.16% | 8.29% | 8.29% |
| **Disco** | **99.94**% | **99.4**% | **98.77**% | **98.04**% | **97.43**% | **96.17**% | **95.99**% | **95.94**% | **95.88**% | **95.84**% |

## O   COMPARISON WITH ADAPTIVE DIVERSITY PROMOTING (ADP) METHOD

Adaptive diversity promoting (ADP) method employs a regularizer while training an ensemble to encourage model diversity. This results in enhancing the robustness for the ensemble because it is difficult to transfer adversarial examples among individual models.

Here, we investigate the performance afforded by the model diversification method against query-based black-box attacks under our proposed model randomization method and compare it with our proposed SVGD+ method for building a set of diverse and well-performing models.

We conduct extensive experiments to compare the robustness of our approach using SVGD+ and ADP under our framework with a configuration of random five out of 10 models against black-box attacks, SQUAREATTACK ($l_2$, $l_\infty$) and SIGNHUNTER ($l_2$, $l_\infty$). The results in Tables 25 and 26 show that our proposed model randomization performs well with the learning objective introduced in Pang et al. (2019). The results also demonstrate that ADP can encourage diversity and Disco(ADP) can achieve comparable robustness to Disco(SVGD+) under *low* perturbation budgets. Under increasing perturbations, models learned with SVGD+ demonstrates improved robustness.

Table 25: **CIFAR−10**. Compare the robustness (higher ↑ is stronger) of our approach using SVGD+ versus ADP against SQUAREATTACK ($l_\infty$) and SIGNHUNTER ($l_\infty$). We randomly select a subset of five models (from ten models).

| Groups | Methods | $l_\infty$ =0.02 | 0.04 | 0.06 | 0.08 | 0.1 |
|---|---|---|---|---|---|---|
| SIGNHUNTER ($l_\infty$) | Disco(ADP) | **99.99**% | **99.59**% | **96.01**% | 89.78% | 82.19% |
| | **Disco(SVGD+)** | 99.97% | 99.95% | 95.56% | **90.71**% | **84.22**% |
| SQUAREATTACK ($l_\infty$) | Disco(ADP) | **99.99**% | 96.31% | 83.99% | 65.28% | 50.41% |
| | **Disco(SVGD+)** | 99.97% | **96.91**% | **86.52**% | **70.22**% | **55.77**% |

Table 26: **CIFAR−10**. Compare the robustness (higher ↑ is stronger) of our approach using SVGD+ versus ADP against (SQUAREATTACK ($l_2$), SIGNHUNTER ($l_2$)). We randomly select a subset of five models (from ten models).

| Groups | Methods | $l_2$ =0.8 | 1.6 | 2.4 | 3.2 | 4.0 |
|---|---|---|---|---|---|---|
| SIGNHUNTER ($l_2$) | Disco(ADP) | **99.98**% | **99.62** % | 97.54% | **94.06**% | 89.07% |
| | **Disco(SVGD+)** | 99.96% | 99.24 % | **97.61**% | 93.63% | **90.24**% |
| SQUAREATTACK ($l_2$) | Disco(ADP) | **99.91**% | **97.12** % | **87.71**% | 76.35% | 63.81% |
| | **Disco(SVGD+)** | 99.56% | 95.62% | 87.07% | **76.50**% | **65.76**% |

Interestingly, from the analysis of clean accuracy drop in Table 27, we can observe the learning objective we introduced in Section 3.4.2 allows Disco(SVGD+) to achieve a lower clean accuracy drop than Disco(ADP).

Table 27: Clean accuracy and clean accuracy drop ($\downarrow \Delta$) comparison between ADP models versus SVGD+ training objective based models on CIFAR-10. *All* represents results from the entire ensemble of models while Disco(.) represents performance under the model randomization configured with five out of 10 models.

| ADP (All) | Disco(ADP) ($\downarrow \Delta$) | SVGD+ (All) | Disco(SVGD+) ($\downarrow \Delta$) |
|-----------|----------------------------------|-------------|-------------------------------------|
| 94.56% | 93.29% ($\downarrow$1.27%) | 93.19% | 92.26% ($\downarrow$**0.93%**) |

