# OpenReview forum: "Can Model Randomization Offer Robustness Against Query-Based Black-Box Attacks?"
_ICLR.cc/2025/Conference — ICLR 2025 Conference Withdrawn Submission_

### Official Review · Reviewer_Scpz · 2024-11-01

**Soundness:** 3
**Presentation:** 4
**Contribution:** 3
**Rating:** 6
**Confidence:** 3

**Summary:**

The paper explores a novel defense mechanism to improve the robustness of deep learning models against query-based black-box adversarial attacks. The authors propose using model randomization as a defense. This obfuscates the relationship between successive responses, thereby hindering the adversary's optimization process for generating adversarial examples. The paper provides both theoretical analyses and empirical results showing that model randomization improves resilience across various attack types and perturbation objectives.

**Strengths:**

1. Well-written and Clear: The paper is clearly articulated, making complex concepts accessible to the reader. The structure of the paper allows for an easy understanding of the problem, solution, and results.
2. Theoretical Justification of Randomization: The paper provides a strong theoretical foundation, demonstrating how model randomization can enhance robustness against adversarial attacks.
3. Rigorous Experiments: The experiments are comprehensive and rigorously conducted, covering multiple types of adversarial attacks (score-based, decision-based) and perturbation objectives (l∞, l2, l0). This enhances the credibility of the paper's claims regarding the method’s efficacy.

**Weaknesses:**

1. Over-Reliance on Test Accuracy: As noted in Appendix C (around line 1020), it appears that model selection is primarily based on test accuracy. (If I have misunderstood this aspect, I would appreciate any clarification. Addressing this concern would greatly encourage me to consider a more favorable evaluation of the paper.)

2. Experiments Limited to Low-Resolution Data: The experiments are conducted on relatively low-resolution datasets like MNIST, CIFAR-10, and STL-10. This limits the generalizability of the results to more complex, high-resolution datasets, which are more common in real-world scenarios.

**Questions:**

1. Model Selection: Could the authors clarify the exact process used for model selection? If test accuracy was indeed a criterion, it would be helpful if the authors could explain why this approach was chosen over using a validation set. Addressing this concern would greatly encourage me to consider a more favorable evaluation of the paper.

2. High-Resolution Data: Have the authors conducted experiments on high-resolution datasets (e.g., ImageNet)? It would be valuable to assess how the proposed defense mechanism performs on more complex, high-dimensional data commonly found in real-world applications.

3. Dataset-Specific Models: The paper uses different models for each dataset, which raises concerns about whether the results are model-specific. Could the authors clarify the reasoning behind selecting different models for each dataset and provide results that demonstrate the method’s effectiveness across a wider range of models? This would help ensure the robustness of the method across various architectures.

4. Incomplete Released Code: Although the paper mentions that the code is available on GitHub, the provided repository does not include the training code, which is crucial for reproducing the results. To ensure full reproducibility and transparency, it would be helpful if the authors could include specific components of the training process in their code release, such as the implementation of the SVGD+ method, the model randomization procedure, and any custom loss functions or optimizers used. Including these details would provide clearer guidance on what is needed for accurate replication of the results.

---

> ### Author Response · Authors · 2024-11-24
> **We clarify model selection follows standard practice. Provide new results for ImageNet on CLIP and confirm full code release at our public repo**
>
> __Q1. Model Selection__
>
> As we aim to design a defense mechanism that can mitigate compromising performance for robustness, while training a set of models together e.g. 10 models, we select a model set that obtains the best clean accuracy. To achieve this, we follow the standard training scheme and choose the model set at the epoch where the best clean accuracy on a test set is achieved.
>
> __Why this approach was chosen over using a validation set?__
>
> A validation set is used to tune hyperparameters (hyperparameter choices) and assess model performance during training. For datasets such as, MNIST, CIFAR-10 and STL-10 define only two standard subsets: i) training; and ii) testing. You could split the training set into training and validation but ultimately, benchmarks report the best performance (accuracy) on the standard testing set. In our study, we used these datasets to train models, and chose models based on their performance on test set. This is a common practice [1, 2]
>
> [1] He, Kaiming, X. Zhang, Shaoqing Ren and Jian Sun. “Deep Residual Learning for Image Recognition.” IEEE Conference on Computer Vision and Pattern Recognition (CVPR) (2016).
>
> [2] Zagoruyko, Sergey and Nikos Komodakis. “Wide Residual Networks.” ArXiv abs/1605.07146 (2016)
>
> __Q2 and Q3. Effectiveness on High-resolution dataset and effectiveness across a wider range of models__
>
> Thank you for this suggestion. We have conducted new experiments to address this concern.
>
> As we discussed with __Reviewer WLVn__, to demonstrate the effectiveness of our defense mechanism on a high-resolution dataset (Imagenet) and effectiveness across models (convolutional and transformer-based networks), we conducted an experiment.
> - For our proposed defense, we fine-tuned a set of five CLIP models with LoRA on ImageNet (The model set achieves 78.05% clean accuracy). At test time, we randomly select two out of five models to make predictions.
> - For RND and RF defenses, we fine-tune the CLIP model (achieve 76.07 % clean accuracy). Then we set hyperparameters such that the clean accuracy drop of these two defenses is around 1%.
> - We randomly selected 100 correctly classified images. The results in Table 4 show that our approach works well on ImageNet and achieves better robustness than RND and RF methods. Thus, our defense is effective across different model types, scales well to large models and high-solution datasets like **Imagenet**.
>
> Table 1: $l_\infty$ objective. Robustness ($\uparrow)$ of different defense methods against SQUAREATTACK with the __ImageNet__ task with CLIP model architecture.
> | Methods | 0.025      | 0.05       | 0.075      | 0.1       |
> | ------- | ---------- | ---------- | ---------- | --------- |
> | RND     | 83.39%     | 61.95%     | 43.37%     | 24.89%    |
> | RF      | 86.45%     | 65.1%      | 51.14%     | 35.83%    |
> | DISCO   | __90.76%__ | __72.51%__ | __56.17%__ | __45.4%__ |
>
> Nevertheless, it is important to highlight:
>
> - First, while our initial experiments were conducted on low-resolution datasets to thoroughly analyze the core principles of DISCO,  the proposed framework remains highly relevant and scalable to high-resolution datasets such as ImageNet.
> - Second, the fundamental mechanisms underlying our defense, model diversity and model randomization, are not inherently tied to a specific-mode architecture. However, due to our limited resources and our main aim being to theoretically and empirically examine the effectiveness of our defense idea, we only chose one network architecture for each dataset. We have addressed this now.
>
> __Q4. Code release__
>
> We want to strongly re-firmly our full commitment to reproducibility and transparency.
>
> A partial code release has been done to allow our defense to be evaluated during the paper submission and review. In the next phase, the complete code in our study will be released. It will include:
>
> - The implementation of the SVGD+ training method
> - The model randomization procedure (it should be ready in the released code but will be updated)
> - Code for custom loss functions.

---

> ### Author Response · Authors · 2024-12-03
> **Please let us know if our answers have provided the clarification you have sought?**
>
> - We thank you for the opportunity to answer the important question you have raised.
> - We stand ready to provide any further clarifications or results.

---

### Official Review · Reviewer_r45L · 2024-11-01

**Soundness:** 3
**Presentation:** 3
**Contribution:** 3
**Rating:** 5
**Confidence:** 3

**Summary:**

This paper investigates model randomization as a defensive strategy against query-based black-box attacks on deep neural networks. The authors argue that these attacks exploit the correlation between successive model responses to queries. By randomizing the model responsible for each response, this correlation can be obscured, thereby making the attack more challenging. A theoretical framework is proposed in which models are sampled from a diverse pool to respond to queries, increasing uncertainty for the attacker and degrading the quality of information that can be extracted from query responses.

To encourage model diversity without compromising individual model performance, the authors introduce a novel learning objective. Extensive empirical evaluations demonstrate that the proposed method, named Disco, substantially enhances model robustness against advanced query-based black-box attacks across various perturbation norms ($\ell_\infty$, $\ell_2$, $\ell_0$) and adaptive attack strategies, while maintaining high accuracy on clean data.

**Strengths:**

This work proposes an interesting insight and analyzes its validity theoretically, then presents supporting evidence experimentally.
Experimental results show that the proposed method is highly effective, maintaining strong robustness without compromising accuracy.

**Weaknesses:**

1. The emphasis on specific architectures, such as Convolutional Neural Networks (CNNs), and small datasets, including MNIST, CIFAR-10, and STL-10, offers valuable insights. However, this focus may restrict the generalizability of the approach's effectiveness across a broader spectrum of models and large data types (such as the ImageNet dataset or the COCO dataset).

2. While the proposed method demonstrates enhanced robustness, could the authors provide an analysis of its computational overhead?  In particular, how do time and resource requirements measure up when comparing inference time and memory usage with existing techniques on a standardized hardware setup?

**Questions:**

1. The abstract mentions that state-of-the-art attacks were employed to evaluate the proposed method; however, current query-based black-box attacks that incorporate prior knowledge, such as PBO [1], have demonstrated significantly better performance than the methods discussed in the paper. It would be beneficial to consider these more advanced techniques in the evaluation, where the souce code of P-BO can be found in https://github.com/machanic/P-BO.

2. The mathematical notation in Section 3.3.1 is ambiguous. For clarity, vector $\mathbf{u}$ should have the same dimension as $\mathbf{x}$ to allow summation in Equations (4), (5), and (6). Similarly, $g(\mathbf{x})$, as the estimated gradient, should align dimensionally with $\mathbf{x}$. In Proposition 1, it is unclear how $g(\mathbf{x})$ is bounded by the constants $a_i$ and $b_i$. If $a_i$ and $b_i$ are vectors, why would the lower bound for $n$ in the conclusion then be represented as a vector?

3. While the theoretical analysis considers two aspects (GRADIENT ESTIMATION ATTACKS and GRADIENT-FREE ATTACKS), some uncertainties persist. For example, as noted in Section 3.3.2, $P(\frac{\tilde{H}(\mathbf{x}, \mathbf{u})}{\hat{H}(\mathbf{x},\mathbf{u})})<0$ represents the probability of misleading an attack direction. However, in practical applications, it may sometimes be necessary to consider the opposite direction due to potential inaccuracies in direction (as seen in methods like SignOPT [2]). Thus, relying solely on the sign may not fully capture the misleading nature of the search direction.

[1] Cheng, S., Miao, Y., Dong, Y., Yang, X., Gao, X.-S., & Zhu, J. Efficient Black-box Adversarial Attacks via Bayesian Optimization Guided by a Function Prior. In *Proceedings of the 41st International Conference on Machine Learning (ICML)*, pp. 8163–8183, 2024.

[2] Minhao Cheng, Simranjit Singh, Pin-Yu Chen, Sijia Liu, and Cho-Jui Hsieh. Sign-OPT: a query-efficient hard-label adversarial attack. In *International Conference on Learning Representations*, pp. 1–16, 2020.

---

> ### Author Response · Authors · 2024-11-23
> **We address all the weaknesses: The method is model-agnostic but we show generality to other architectures and results for overhead**
>
> ***Thank you for your initial evaluation of our work and great feedback to improve our work!***
>
> __Q1. The generalizability of the approach's effectiveness across a broader spectrum of models and large data types__
>
> We agree and are happy to report new results to address this concern.
> - As we discussed with **Reviewer WLVn**, we fine-tuned a CLIP model to build an ensemble of 5 particles (models). This network is very different from VGG and ResNet architectures.
> - For comparison, we also fine-tuned a single CLIP model for RND and RF defenses, achieving a clean accuracy of 76.07% for the single model. Noise was selected to maintain a clean accuracy drop of approximately 1% as in the prior work.
> - For this experiment, we randomly selected 100 correctly classified images.
> - The results in Table 4 show that DISCO achieves higher robustness compared to RND and RF methods. Thus, DISCO is effective across different model types, scales well to large models and generalises to challenging datasets like **Imagenet**.
>
> Table 1: $l_\infty$ objective. Robustness ($\uparrow$) of different defense methods against SQUAREATTACK with the __ImageNet__ task with CLIP architecture [2]
> | Methods | 0.025      | 0.05       | 0.075      | 0.1       |
> | ------- | ---------- | ---------- | ---------- | --------- |
> | RND     | 83.39%     | 61.95%     | 43.37%     | 24.89%    |
> | RF      | 86.45%     | 65.1%      | 51.14%     | 35.83%    |
> | DISCO   | __90.76%__ | __72.51%__ | __56.17%__ | __45.4%__ |
>
> __Q2. An analysis of its computational overhead__
>
> We recognize that storing and querying multiple models can lead to increased computational and storage demands.
>
> To address this, we did two things.
>
> 1. We conducted a detailed comparison of storage requirements and inference time across DISCO, RND, and RF, as presented in Tables 2 and 3.
>     - Notably, both RND and RF rely on a single model, resulting in identical storage requirements.
>     - In contrast, DISCO uses a set of 40 models for MNIST and 10 models for CIFAR-10/STL-10. For the results reported in Table 2, we processed 1,000 queries and calculated the average inference time to provide a fair and comprehensive evaluation.
> 2. We demonstrate that the overhead needed to achieve the high robustness levels can be dramatically minimized by using recent advances to reduce the cost of building a set of networks based on CLIP [1], such as LoRA [2] and the recent study [3].
>     - Please see our discussion with __Reviewer YFcN__ and the results reported in ***Table 4*** there.
>
> _Table 2:_ Storage Consumption of models trained on different datasets between RND, RF vs DISCO.
> | Datasets | RND & RF | DISCO |
> | -------- | -------- | ----- |
> | MNIST    | 1.3 MB    | 145 M |
> | CIFAR-10 | 57 MB     | 1.7 G |
> | STL-10   | 43 MB     | 1.3 G |
>
> Table 3: Inference time (per query) of undefended vs defended (RND, RF vs DISCO) models on different datasets.
> | Datasets |  Undefended   | RND | RF  | DISCO |
> |---| --- | --- | --- | ----- |
> |MNIST|10.17 ms|12.14 ms   |  12.53 ms   |   15.12 ms    |
> |CIFAR-10|10.56 ms|12.61 ms   |  12.92 ms   |   20.62 ms    |
> |STL-10|11.26 ms|13.12 ms   |   13.48 ms  |   24.85 ms   |
>
> [1] Radford, Alec, Jong Wook Kim, Chris Hallacy, Aditya Ramesh, Gabriel Goh, Sandhini Agarwal, Girish Sastry, Amanda Askell, Pamela Mishkin, Jack Clark, Gretchen Krueger and Ilya Sutskever. “Learning Transferable Visual Models From Natural Language Supervision.” ICLR, 2021.
>
> [2] Hu, E. J.; yelong shen; Wallis, P.; Allen-Zhu, Z.; Li, Y.; Wang, S.; Wang, L.; and Chen, W. LoRA: Low-Rank Adaptation of Large Language Models. ICLR, 2022.
>
> [3] Doan, Bao Gia, Afshar Shamsi, Xiao-Yu Guo, Arash Mohammadi, Hamid Alinejad-Rokny, Dino Sejdinovic, Damith Chinthana Ranasinghe and Ehsan Abbasnejad. “Bayesian Low-Rank LeArning (Bella): A Practical Approach to Bayesian Neural Networks.” ArXiv (2024).

---

> ### Author Response · Authors · 2024-11-24
> **Questions to Authors: We report new results for P-BO Attack & Provide clarifications**
>
> __Q3. The performance of defense mechanisms under recent advanced attack P-BO__
>
> Thank you for the suggestion. We have been able to test with the attack recommended by the Reviewer. We report the new result in Table 4 below.
>
> - Ours (DISCO) is demonstrably more robust.
> - We are in the process of updating our paper with this new result.
>
> Table 4: $l_\infty$ objective. Robustness of different defense methods against P-BO attack with the __CIFAR-10__ task.
> |Methods|0.02|0.04|0.06|0.08|0.1|
> |---|---|---|---|---|---|
> |RND|70.33%|31.47%|15.75%|7.23%|6.65%|
> |RF|66.43%|28.04%|13.67%|8.34%| 6.21%|
> |DISCO|__79.98%__|__47.94%__|__29.8%__|__18.08%__|__12.16%__|
>
> __Q4. Capture the misleading nature of the search direction__
>
> Thank you for your constructive feedback. Let us explain further here:
>
> - In Section 3.3.2, line 267-269, we use $\frac{\tilde{H}(x,u)}{\hat{H}(x,u)}<0$ to represent the *mismatch* between the attack direction from a random subset of models versus that generated using the entire set of models (effectively a single model).
> - This condition specifically reflects opposite directions as it relies on the *different* sign of search direction. Therefore, line 269, $P(\frac{\tilde{H}(x,u)}{\hat{H}(x,u)}<0)$ represents the probability of misleading an attack direction.
> - This mislead direction is not limited to arbitrary discrepancies but explicitly incorporates the notion of opposing directions. To this end, we believe that our theoretical analysis in Section 3.3.2 on misleading a search direction is able to fully capture the misleading nature of the search direction described by the Reviewer.
>
> We really appreciate the observation and will clarify this point further in the manuscript to ensure its practical implications are well-understood.

---

> > ### Comment · Reviewer_r45L · 2024-11-25
> >
> > Thank you for your response.
> >
> > Unfortunately, my concern regarding Question 2—the ambiguity in the mathematical notation in Section 3.3.1—remains unaddressed. This ambiguity suggests there may be a mistake in the theoretical analysis presented. As a result of this unresolved issue, I have decreased my score to 5.

---

> ### Author Response · Authors · 2024-11-25
> **Reviewer r45L, We just wanted to let you know, we have now finished responding to Question 2.**
>
> ***Thank you for careful review. We really appreciate the opportunity to clarify this important point. We are sorry this reply is late, we were working through all of our new results first.***
>
> We will go through the answer, step by step.
>
> 1. __Dimension of $u$, $x$ and $g(x)$__: You are correct that the vector $u$ and esitmated gradient $g(x)$ should have the same dimension as $x$ to ensure proper summation in Equations(4), (5), and (6). We will revise the notation to explicitly state that $u \in R^d$, $x\in R^d$ and $g(x) \in R^d$.
> 2. __The bound of g(x)__: As $g(x)$ is a vector, the bound of $g(x)$ can be defined as the bound for each element (each dimension) of $g(x)$. To eliminate ambiguity, we will formally define the bound of $g(x)$ as follow:
> $$a_{i}^j \leq g\(x\)\_{i}^j\leq b_{i}^j$$
>    - where $i=1,..., n$, $n$ is the number of different gradient estimators $g(.)$, $j=1,..., d$ for each dimension, $d$ is the number of elements (dimensions) of $g(x)_i$.
>
>    To simplify the notation, we drop $i$ for gradient estimator $g(x)_i$ to have $g(x)$.
>
>  3. __The lower bound for n__: We are sorry for the lack of clarity here and describing the bound for each scalar element without expclicity describing this clearly. To address the confusion, we provide an update for equation (7) as follow:
>    - We define $A^j$ as $|\bar{g}(x)^j - \hat{G}(x)^j|\geq \Delta$
>    - According to the Hoeffding's inequality and employing a union bound over all $d$ dimensions to bound the probability of deviation in any component, we have:
>  $$P(\cup_{j=1}^{d}A^{j})\leq \sum_{j=1}^{d} P(A^j)=\sum_{j=1}^{d}2\exp{\Big(-\frac{2n^2\Delta^2}{\sum_{i=1}^{n}(a_{i}^j-b_{i}^j)^2}\Big)}.$$
>  This term can further be upper bounded by considering the fact that $\exp(-x)$ is monotonically decreasing, we know for any $j$:
>  $$\exp{\Big(-\frac{2n^2\Delta^2}{\sum_{i=1}^{n}(a_{i}^j-b_{i}^j)^2}\Big)}\leq \exp{\Big(-\frac{2n^2\Delta^2}{\sum_{i=1}^{n}[\max_j(a_{i}^j-b_{i}^j)^2]}\Big)}$$
>  Therefore, we have:
>  $$P(\cup_{j=1}^{d}A^{j}) \leq \sum_{j=1}^{d}2\exp{\Big(-\frac{2n^2\Delta^2}{\sum_{i=1}^{n}(a_{i}^j-b_{i}^j)^2}\Big)} \leq 2d\exp{\Big(-\frac{2n^2\Delta^2}{\sum_{i=1}^{n}[\max_j(b_{i}^j-a_{i}^j)]^2}\Big)}$$
> To achieve low margin error $\Delta$ with the desired confidence level $1 − \delta$ and the given bound as above, we set the right-hand side of the inequality smaller than $\delta$ and solve for n as the following:
>
>
>
> $$
> \begin{equation}
>  2d\exp{\Big(-\frac{2n^2\Delta^2}{\sum_{i=1}^{n}[\max_j(b_{i}^j-a_{i}^j)]^2}\Big)}\leq\delta
> \end{equation}
> $$
> $$
> \begin{equation}
> -\frac{2n^2\Delta^2}{\sum_{i=1}^{n}[\max_j(b_{i}^j-a_{i}^j)]^2}\leq \log\frac{\delta}{2d}
> \end{equation}
> $$
> $$
> \begin{equation}
>  \frac{2n^2\Delta^2}{\sum_{i=1}^{n}[\max_j(b_{i}^j-a_{i}^j)]^2}\geq \log\frac{2d}{\delta}
> \end{equation}
> $$
> $$
> \begin{equation}
>  n^2\geq \frac{\log\frac{2d}{\delta}\sum_{i=1}^{n}[\max_j(b_{i}^j-a_{i}^j)]^2}{2\Delta^2}
> \end{equation}
> $$
> $$
> \begin{equation}
>  n\geq \sqrt{\frac{\log\frac{2d}{\delta}\sum_{i=1}^{n}[\max_j(b_{i}^j-a_{i}^j)]^2}{2\Delta^2}}
> \end{equation}
> $$
>
> We hope that our response addressed the concerns of the reviewer. We're happy to promptly answer any additional questions.

---

> ### Comment · Reviewer_r45L · 2024-11-26
> **Thank you for your response**
>
> Thank you for your detailed response. While I consider the mathematical derivation in your last reply to be correct, I believe **the primary concern with this paper lies in the significant computational and resource costs associated with training**, as highlighted in comments of Reviewers WLVn and whE1. For example, using ImageNet as a case study, training an ensemble of multiple models would require extensive time and GPU resources, leading to increased carbon emissions and financial burden. Would it be possible for your method to take advantage of pre-trained models, such as those provided by the `timm` library for the ImageNet dataset?

---

> ### Author Response · Authors · 2024-11-27
> **Response to Further Questions (Yes we have now used a pre-trained model, cost overhead is <0.38% per addition of a model)**
>
> **Derivation is fixed** *While I consider the mathematical derivation in your last reply to be correct...*
>
> *Response:*
>
> Thank you for allowing us to answer your question and address the basis for lowering your evaluation of our work.
>
> ---
>
> We are happy to answer the new feedback and we appreciate the opportunity to engage with the Reviewer to add your concerns.
>
> **Q2** *I believe the primary concern with this paper lies in the significant computational and resource costs associated with training. Would it be possible for your method to take advantage of pre-trained models, such as those provided by the timm library for the ImageNet dataset?*
>
> *Response:*
>
> **Yes**,  our method can indeed leverage pre-trained models, such as those available in the `timm` library or other publicly available repositories. In fact, we have done this with a ***pre-trained CLIP*** model from `https://github.com/mlfoundations/open_clip` to demonstrate practicability and a working defense for a model suitable for serious applications.
>
> ***OpneCLIP*** is an open-source implementation of ***OpenAI's CLIP (Contrastive Language-Image Pre-training)***
>
> - We stated with a ***pre-trained CLIP*** (to put CLIP into perspective, please note that ***RestNet50 has approximately 25 million parameters compared to 114 million we deal with in CLIP***).
> - We then trained an ensemble of 5 using the methods in [3].
>     -  The model set achieves approximately 78% clean accuracy.
>     -  During the inference phase, two out of five were randomly selected from the trained set.
>     -  Two out of five has a clean test accuracy of approximately 77%
> - We also train a single CLIP model for use with RND and RF
>     - The single model achieves 76.07% clean accuracy
>     - We add noise, so the clean accuracy drop values are as in RF (ICLR'24), approximately 1% and comparable to ours.
> - Table 4 shows that the significant reduction in overhead we achieve.
> - Table 5 shows that compared to current state-of-the-art we outperform with a margin of up to 9.57%
>
> _Table 4:_ Trainable Parameters and Storage Consumption of a Single CLIP and a set of five CLIP models ***we trained*** to implement LoRA(DISCO).
> | Models|***Single*** CLIP|The set of 5 CLIP models using LoRA|
> | -------------------- | -----------|------|
> | Trainable Parameters |114 M| 1.84  M  (1.6% incrase, ***0.32% per model***)|
> | Storage Consumption  |433 MB|  439 MB (1.35% increase, ***0.28% per model***)|
>
> _Table 5:_ $l_\infty$ objective. Robustness ($\uparrow)$ of different defense methods against SQUAREATTACK with the __ImageNet__ task task with __CLIP__ model architecture [3] (For details on the experiment, please see response to Reviewer).
> | Methods | 0.025      | 0.05       | 0.075      | 0.1       |
> | ------- | ---------- | ---------- | ---------- | --------- |
> | RND     | 83.39%     | 61.95%     | 43.37%     | 24.89%    |
> | RF      | 86.45%     | 65.1%      | 51.14%     | 35.83%    |
> | DISCO   | __90.76%__ | __72.51%__ | __56.17%__ | __45.4%__ |
> | DISCO Improvement (vs. Next best)|4.31%|7.41%|5.03%|9.57%|
>
> What we are proposing are ***marginal cost increases*** to achieve significant improvements in robustness.
>
> - Effectively we are saying <1.6% increase in overhead can yeild 4.31 to 9.57% better robustness on a large-scale network of practical significance
> - Now adding a model incurs **<0.32%** overhead in terms or trainable parameters or storage.
>
>
> We really hope the results provides the assurances sought by the Reviewer that our methods is:
> - Robust and
> - Practical
>
> We hope the reviewer can appreciate the question we posed in the paper:
> - CAN MODEL RANDOMIZATION OFFER ROBUSTNESS AGAINST QUERY-BASED BLACK-BOX ATTACKS?
>   - We believe the answer is now, yes, irrevocably.
>   - In addition to understating what such a method can offer, and the theoretical analysis, we have now shown it can be of practical significance.
>   - One paper can't solve all problems, but we have certainly worked hard at making our theoretical work stick.
>
> We sincerely thank the Reviewer for all their efforts to help us improve our work and its presentation.

---

### Official Review · Reviewer_9act · 2024-11-02

**Soundness:** 2
**Presentation:** 3
**Contribution:** 3
**Rating:** 6
**Confidence:** 2

**Summary:**

This paper explores whether model randomization can enhance robustness against query-based black-box attacks. Traditional defenses involve random noise injections, but this study proposes a novel defense: generating responses by sampling from an ensemble of diverse models. Theoretical analysis and extensive empirical tests across various attacks and perturbation metrics validate the efficacy of this strategy, achieving robustness with minimal performance compromise.

**Strengths:**

1. Good writing with clear motivation and insights.
2. Authors provide a strong theoretical foundation, with proofs demonstrating how randomization increases the difficulty of gradient estimation and search-based attacks.
3. By selecting models with a diversity-promoting training objective, the paper manages to keep clean accuracy relatively stable.

**Weaknesses:**

1. How does increasing model diversity impact clean accuracy, and is there a systematic way to balance these two metrics?
2. Disco tests VGG and ResNet architectures, I think it would be useful to know whether this approach is effective across more different model types.
3. Although Disco's results on MNIST, CIFAR-10, and STL-10 are promising, these are relatively small datasets. It would strengthen the paper to see how this defense performs on more challenging datasets, like ImageNet.
4. In the paper, authors mention that they use a larger number of models (40) for the MNIST task and a lower number of models (10) for high-resolution CIFAR-10 and STL-10 tasks. Have the authors tested the impact of different ensemble sizes, and is there an optimal balance between model count and robustness?

**Questions:**

See above.

---

> ### Author Response · Authors · 2024-11-24
> **We address the concerns about balancing diversity and accuracy, effectiveness across model types and datasets and the impact of different sizes (Appendix J)**
>
> ***Thank you for the valuable comments. This has helped us improve our work***
>
> __Q1. A systematic way to balance model diversity and clean accuracy__
>
> Thank you for raising an important point. This is studied in depth in deep Bayesian neural networks.
> - First, as expected, increasing model diversity alone will come at the expense of model performance.
> - The SVGD method provides a systematic method to control this balance. A model trainer can determine the empasis placed on diversity vs. model performance by selecting a suitable $\gamma$ paramter for Equation (10) as we mentioned in Line 332.
>
> __Q2 and Q3. Effectiveness across different model types and datasets__
>
> We agree and are happy to report new results to address this concern.
> - As we discussed with **Reviewer WLVn**, we fine-tuned a pre-trained CLIP model to build an ensemble of 5 models, which is different from VGG and ResNet architectures.
> - Importantly, we addressed the computational demands of the approach by employing LoRA to build a set of five models from a single pre-trained CLIP model to achieve approximately 78% clean accuracy. During the inference phase, two out of five were randomly selected from the trained set.
> - For comparison, we also fine-tuned a single CLIP model for RND and RF defenses, achieving a clean accuracy of 76.07% (noise was selected to maintain a clean accuracy drop of approximately 1% as in the prior works). For this experiment, we randomly selected 100 correctly classified images.
> - The results in Table 1 show that DISCO achieves higher robustness compared to RND and RF methods. Thus, DISCO is effective across different model types, scales well to large models, and works well on challenging datasets like **Imagenet**.
>
> Table 1: $l_\infty$ objective. Robustness of different defense methods against SQUAREATTACK with the __ImageNet__ task.
> | Methods | 0.025      | 0.05       | 0.075      | 0.1       |
> | ------- | ---------- | ---------- | ---------- | --------- |
> | RND     | 83.39%     | 61.95%     | 43.37%     | 24.89%    |
> | RF      | 86.45%     | 65.1%      | 51.14%     | 35.83%    |
> | DISCO   | __90.76%__ | __72.51%__ | __56.17%__ | __45.4%__ |
>
> __Q4. The impact of different sizes of the model set__
>
> Thank you for this insightful question. Indeed, we used the MNIST to enable us to train large sets of models to investigate the question posed by the Reviewer.
> - Our extensive results with MNIST show that as long as we can maintain model diversity and performance, having a large pool of models to select from is more robust. To demonstrate, we have *extracted* the results in ***Appendix J *** (in the revised manuscript) and shown the impact of selecting 1 from 10, 20 and 40 models under a strong attack budget (harder to defend against). Please see results in ***Table 2*** below. This demonstrates that having a large pool is beneficial.
> - Recall, robustness of a model is correlated with the number of queries needed to mount a successful attack. Then, from our theoretical analysis, robustness relies on the output score variance from our model randomisation approach.
>   - So, as we discussed with Reviewer ***WLVn*** in Q3, increasing the ensemble size leads to improved robustness due to larger variance in model outputs.
>
> Table 2: $l_2$ objective. The robustness of DISCO against SQUAREATTACK at the strong attack budget 4.0 when sampling one out of different sizes of model sets.
> | Random Selection | Accuracy |
> | ---------------- | -------- |
> | 1 out of 10      |     80.0%     |
> | 1 out of 20      |     83.5%     |
> | 1 out of 40      |     88.2%     |

---

### Official Review · Reviewer_whE1 · 2024-11-04

**Soundness:** 3
**Presentation:** 3
**Contribution:** 2
**Rating:** 6
**Confidence:** 2

**Summary:**

The paper proposes a defense mechanism called Disco against query-based black-box adversarial attacks by randomizing model responses. The core idea is that model randomization—drawing models from a pool of well-trained diverse models for each query—can obfuscate the consistent feedback adversarial attacks depend on. By breaking the static relationship between query responses, this method aims to degrade an attacker's ability to generate effective adversarial examples. The authors conduct extensive theoretical and empirical evaluations, examining attacks across multiple threat models and perturbation types, and find that Disco is shown to outperform existing defenses against black-box attacks.

**Strengths:**

- The proposed defense is straightforward and simple to implement, making it a practical solution for enhancing model robustness against adversarial attacks.
- Theoretical analysis thoroughly supports the effectiveness of model randomization in the proposed defense.
- The paper conducts an extensive evaluation across various threat models and perturbation types, providing a robust assessment of the defense's performance.

**Weaknesses:**

- The approach requires substantial resources, as multiple models must be trained during the training phase. Additionally, switching between models during inference could increase response time and demand more server resources.
-  Models trained on the same dataset may share similar adversarial boundaries, posing a risk that attackers could exploit if the decision boundaries are too alike across models in the pool.

**Questions:**

1. Could you give some solution about how to reduce the resource overhead in both training and inference phase?
2. Could you explain how possible or impossible that all models in the pool share a similar adversarial boundaries?

---

> ### Author Response · Authors · 2024-11-23
> **We provide a solution to reduce overhead. We explain how our learning objective (Equation 10) addresses similar adversarial boundaries**
>
> ***Thank you for all the valuable comments!***
>
> __Q1. Solutions to reduce the resource overhead in both the training and inference phases__
>
> To reduce resource overhead in both the training and inference phases, we can:
>
> - Leverage LoRA as we have already demonstrated. Simply, we can incorporate LoRA to build an ensemble from a single pre-trained model and employ SVGD to push the parameters of LoRA apart. This can significantly reduce the need to train and store multiple large models while maintaining diversity. LoRA modifies only a small subset of parameters, which can make both training and inference more resource-efficient.
> - We now report results from using LoRA for a CLIP model for the **ImageNet** task. Please see __Reviewer YFcN, Table 4__.
> - Notably, one could also explore other efficient diversity-promoting schemes. To illustrate, we can dynamically generate diversity through parameter perturbation or layer-wise modularity, which could reduce storage and computational costs.
>
> __Q2. Share similar adversarial boundaries__
>
> Thank you for the interesting question. Indeed, this is why we:
>
> 1. Select a random model and
> 2. Build models that learn *different* representations (Objective in Equation 10)
>
> This reduces the possibility of adversarial boundaries being the same. Effectively, during training, the objective in (10) pushes model parameters apart in the parameter space, fostering unique representations across models. This results in individual models with different decision boundaries.
>
> Indeed, ***if*** the models were not *diverse* (for example, we did not use the objective in 10), we can expect the models in the pool to possibly share adversarial boundaries.

---

> > ### Comment · Reviewer_whE1 · 2024-11-25
> >
> > Thanks for the clarification. It addresses my concerns. I would like to keep my score.

---

> ### Author Response · Authors · 2024-11-25
> **Thank you for your valuable feedback Reviewer whE1! We will share an updated PDF.**
>
> - We appreciate the Reviewer for checking back our responses.
> - We are updating the PDF to reflect the discussion we have above.

---

### Official Review · Reviewer_YFcN · 2024-11-04

**Soundness:** 3
**Presentation:** 3
**Contribution:** 3
**Rating:** 6
**Confidence:** 5

**Summary:**

This paper proposes to defend against query-based adversarial attacks by randomization. Existing defenses use this principle by injecting a random noise into the input, feature, or parameters of the model. On the other hand, this work suggests creating many diverse models and randomly ensembling their predictions to fool the attacker. Their method, named Disco, employs a Bayesian framework to learn a diverse set of models. To handle the accuracy degradation, this work proposes a novel asymmetric training objective that forces each model to perform well. The paper also provides a theoretical analysis showing that randomly ensembling significantly increases the number of queries for a successful attack. Experimental results demonstrate the effectiveness of Disco against score-based attacks, decision-based attacks, and adaptive attacks.

**Strengths:**

- The paper provides a novel approach for randomized defense against query-based attacks. Instead of fixing the model and injecting random noise, they suggest creating many diverse models and randomly selecting some of them to predict.
- The paper proposes a novel training objective to avoid an accuracy drop in each model.
- Theoretical analysis shows that randomly ensembling is effective against score-based attacks.
- Experimental results for a wide range of attacks demonstrate that Disco safeguards the model against query-based attacks.

**Weaknesses:**

- The main disadvantage of this approach is that we need to store a set of models instead of one. Ensembling also requires querying many models during inference, which incurs significant computational costs. This problem is even more severe when we deploy large-scale models in practice. An analysis of the training time, storage, and inference time of Disco compared to other defenses such as RND and RF could be helpful.
- The experiments are conducted on low-resolution datasets only.
- The architecture of the target model in the experiments is also quite limited. There is only one model for each dataset and they are all convolutional networks.
- The paper does not explain why the proposed objective helps each individual model perform well. Can we instead randomly sample a subset of model when training?

**Questions:**

- What is the training time, inference time, and storage of Disco?
- Is Disco still effective on high-resolution datasets such as Imagenet?
- What is the performance of Disco on other types of architectures such as transformers?

---

> ### Author Response · Authors · 2024-11-23
> **We addressed all of the Weaknesses  and Questions. We included new results on ImageNet with OpenCLIP**
>
> __Q1. An analysis of the training time, storage, and inference time of DISCO.__
>
> Indeed, as we discussed with Reviewer **WLVn**, there is __no free lunch__.
> - We report comprehensive comparisons of training time, storage and inference time of Disco, RND and RF in Tables 1, 2 and 3. We note that, RND and RF use a single model, the training time and storage for both of them are the same while DISCO uses 40 models for MNIST and 10 models for CIFAR-10/STL-10.
> - For the results in Table 3, we ran 1000 queries and calculated the average inference time.
>
> Table 1: Training time of models trained on different datasets between RND, RF vs DISCO.
> |Datasets|RND & RF|DISCO|
> |---|---|---|
> |MNIST|~0.5 hr|~12.5 hrs|
> |CIFAR-10|~1.5 hr|~72.0 hrs|
> |STL-10|~1.2 hr|~60.0 hrs|
>
> Table 2: Storage Consumption of models trained on different datasets between RND, RF vs DISCO.
> |Datasets|RND & RF|DISCO|
> |---|---|---|
> |MNIST|1.3 MB|145 M|
> |CIFAR-10|57 MB|1.7 G|
> |STL-10|43 MB|1.3 G|
>
> Table 3: Inference time (per query) of undefended vs defended (RND, RF vs DISCO) models on different datasets.
> |Datasets|Undefended|RND|RF|DISCO|
> |---|---|---|---|---|
> |MNIST|10.17 ms|12.14 ms|12.53 ms|15.12 ms|
> |CIFAR-10|10.56 ms|12.61 ms|12.92 ms|20.62 ms|
> |STL-10|11.26 ms|13.12 ms|13.48 ms|24.85 ms|
>
> ***Notably the results for DISCO inference times can easily be improved with parallelisation of the inference pipeline to nearly match a RND and RF, which we have not done***.
>
> __Q2. Effectiveness on high-resolution datasets such as ImageNet__
>
> We theoretically investigated a new method. So, we used relatively low resolution datasets to be able to complete the significant number of experiments needed to thoroughly analyze the core principles of DISCO.
>
> But our method ***is*** effective on high-resolution datasets. We can demonstrate this now with new results.
> - Inspired by recent work to scale ensembling to large-scale models in [1], we fine-tuned a pre-trained model.
> - We used the large-scale OpenCLIP [2] model with LoRA [3] on ImageNet to build a sample of 5 models, achieving approximately 78% clean accuracy on the test set for the ensemble. We used a random selection of two out of five models in our method, achieving approximately 77% clean accuracy on the test set (1% drop).
> - For RND and RF defenses, we fine-tune a single OpenCLIP model to achieve 76.07% clean accuracy and, for a fair comparison, choose hyperparameters such that the clean accuracy drop of these two defenses is also around 1%.
> - In this experiment, due to limited time and computational effort to run attacks, we randomly selected 100 correctly classified images for attacks.
> - The results in **Table 4** below show that our approach works well on ImageNet and achieves better robustness than RND and RF methods.
>
> Table 4: $l_\infty$ objective. Robustness ($\uparrow)$ of different defense methods against SQUAREATTACK with the __ImageNet__ task with *OpenCLIP* model architecture.
> |Methods|0.025|0.05|0.075|0.1|
> |---|---|---|---|---|
> | RND| 83.39%|61.95%|43.37%|24.89%|
> | RF| 86.45%|65.1%|51.14%|35.83%|
> | DISCO| __90.76%__ | __72.51%__ | __56.17%__ | __45.4%__ |
>
> __Q3. Effectiveness with different network architectures__
>
> We understand and appreciate your concern. Let us explain why and then address it.
>
> - Our theoretical work is independent of model architectures.
> - Then, when we empirically examined the effectiveness of our formulation to improve robustness against query-based black-box attacks, we prioritized using convolutional architectures, they are ubiquitous and simple to work with.
> - However, to address your concern, we have conducted comprehensive experiments on ***ImageNet*** with ***OpenCLIP*** based on recent work in [1].
>     - The results in Table 4 (above) show that our proposed method is more robust than RND and RF and works well with transformer architectures (a non-convolutional network).
>     - We hope the generality shown in the results addresses the Reviewer's concern.
>
> __Q4.  Explanation on how the proposed objective helps each individual model perform well__
>
> Thank you for asking. Let us explain why here, while we improve the explanation in the paper.
>
> - Minimizing the loss over an average of logits for a subset of model faces the same problem we tried to address. Because it promotes strong ensemble performance and does not guarantee that each individual model will perform well.
> - Individual model performance, as we mentioned in **Section 3.4.2**, is very important to ensure a minimal performance degradation for our defense. Because we want any randomly selected model to be well-performing.
> - To this end, the proposed objective, through the joint training process, promotes diversity among models and ensures *each* individual model maintains strong performance.
> - Importantly, we emperically show in ***Appendix C, Table 7***, the proposed objective helps obtain well-performing individual models while achieving diversity, concurrently.

---

> ### Author Response · Authors · 2024-11-23
> **Answers to Questions**
>
> ***Thank you for all your feedback. We have answered all the questions. Please let us know if you need further clarifications***
>
> - What is the training time, inference time, and storage of Disco?
>
> *Response: Plese refer to our answer for **Question 1** above in addressing the Weaknesses in the paper.*
>
> ---
>
> - Is Disco still effective on high-resolution datasets such as Imagenet?
>
> *Response:* Yes it is. Performs better than others.
>
>   - Please refer to our answer for **Question 2** above in addressing the Weaknesses in the paper (Table 4).
>   - Please see our more detailed response, including cost overheads, to [___Reviewer r45L___] https://openreview.net/forum?id=DpnY7VOktT&noteId=xFNcPPYEjd (Table 4 & Table 5)
>
> ---
>
> - What is the performance of Disco on other types of architectures such as transformers?
>
> *Response: Please refer to our answer for **Question 3** above in addressing the Weaknesses in the paper.*
>
> ---
>
> ***We appreciate your help in improving our work!*** Please let us know if you have any further question, we will reply to them promptly.

---

> > ### Comment · Reviewer_YFcN · 2024-11-30
> >
> > Thank you for your comment. I believe including the discussion on CLIP and Imagenet would improve the paper. I still have some concerns as follows
> >
> > **Q1.1:** As mentioned in Sec. 4, the method randomly selects a subset of 5 models to make predictions. Thus, I'd expect the inference to be 5x of the base model. How do you achieve ~1.5x inference time in the case of MNIST and ~2x inference times on CIFAR10 and STL-10?
> >
> > **Q1.2:** In the response to reviewer whE1, you recommend leveraging LoRA to mitigate the number of trainable parameters and storage. However, it is only applicable in the fine-tuning setting. Do you have any suggestions for the scenario where we need to train a model from scratch?
> >
> > **Q4:** If optimizing a subset of models has the same effect as ensembling, can we substitute the first term in (11) with a random subset of models similar to the inference step to reduce the resources for backpropagating every model? Since this is an extended discussion, I do not expect to see experimental results but just want to see the reason for not doing that from the beginning.

---

> ### Author Response · Authors · 2024-12-03
> **We addressed all the new questions. We clarify inference time. Answer additional question on training from scratch**
>
> __Q1.1: Inference Time__
>
> Thank you for pointing this out. In our approach to measure inference time, the measurement and calculation included the first few warm-up runs of a model. These warm-up runs are slower due to various factors like *cache warming* and *JIT compilation* [1]. Effectively, the previously reported inference times, including warm-up runs, significantly diluted the actual inference time, which should be in the *micro second* scale. Additionally, while measuring the inference time, there were other workloads (multiple programs/codes) running together in the same server used for producing results for the rebuttal. This seemed to have affected the measurements.
>
> Therefore, we can kindly update the Reviewer with the following we now measured. We now use a 'quiet' machine and average just the inference time, where the model is in a state ready to receive inputs from an end-user via an API.
>
> _Table 1:_ Inference time (per query) of a single vs a set of models on different tasks.
> | Datasets | Single | A subset of five out of 10 Models (DISCO)    |
> | -------- | ---------- | -------- |
> | MNIST    | ~0.7 us   | ~3.8 us |
> | CIFAR-10 | ~1.9 us   | ~9.8 us |
> | STL-10   | ~2.5 us   | ~12.8 us |
>
> [1] https://medium.com/@MarkAiCode/mastering-pytorch-inference-time-measurement-22da0eaebab7
>
> ---
>
> __Q1.2: In the response to reviewer whE1, you recommend leveraging LoRA to mitigate the number of trainable parameters and storage. However, it is only applicable in the fine-tuning setting. Do you have any suggestions for the scenario where we need to train a model from scratch?__
>
> Yes we do. But, we just want to highlight a few things in case this is lost or forgotten as we now start to move away from the main thesis for our work, to *investigate* and *understand* a new idea both theoretically and empirically. Indeed, our work confirms model randomization ***does offer*** robustness against query-based black-box attacks.
>
> So, let's now answer the question. First, we want to *thank you* for this valuable feedback and engaging with us! It helps us consider and address a broader range of practical considerations to ensure a broader applicability of our framework, ***DISCO***.
>
> In addition to supporting the common practice of using pre-trained models, there are strategies that can help mitigate the overhead associated with training models from scratch, one simple strategy can simply build upon our approach for reducing the costs when using pre-trained models. To elaborate:
>
> - One promising approach is to train a single model to obtain a good pre-trained model in the *first* stage.
> - Then employ an ensemble of LoRA to fine-tune the good pre-trained model to achieve a diverse set of models as we demonstrate with CLIP on ImageNet.
>   - The tuning stage learning objective can still employ our Equation (10) and (11).
> - The first stage can use a portion of the training dataset or another dataset if a datset of sufficient size to train a well performing model from scratch is not available.

---

> ### Author Response · Authors · 2024-12-03
> **We answer additional question on, what if random subset for training is used?**
>
> __Q4: Can we substitute the first term in (11) with a random subset of models?__
>
> Thank you for this interesting question. Certainly food for thought. So let's investigate:
>
> First, let's recall, what we have found is that Equation (10) as a learning objective for an ensemble is very crucial for model diversity (please see our ***Hypothesis 2***, *line 198* in updated paper). Then, Equation (11) addresses the two questions we posed about how to achieve diversity while needing to reduce the ***asymmetry*** in performance between models. Reducing the asymmetry is important, because we want any random combination of models to perform well. This mitigates the sacrifice in performance associated with devising means for achieving robustness. So, Equation (10) for diversity, Equation (11) to promote individual model performance.
>
> - As our ablation study results in ***Appendix D*** shows, *without* Sample Loss, the individual particle (model) performance can be poor, more significantly, the performance we can expect from a model randomization method is also poor (Please see ***Table 7***, ***Appendix D***)
> - As our results in ***Figure 3*** shows, without SVGD (see results for *Ensembles*, blue color bar), the diversity among the learned representations is low. This is the reason that SVGD+ (learning with Equations 10 & 11), lead to the best robustness results (Please see ***Table 13***, ***14***, ***15*** and ***16*** in ***Appendix I*** where we compare *DISCO* method with *Ensembles*, *DivDis*, and *DivReg* ensembling methods with other diversity objectives).
>
> Then, to be clear, in using LoRA, the idea is not just to optimize a set of models but to use is with Equation (10) and (11).
>
> Now, to answer the question, substituting the first term in equation (11) with a random subset of models is an interesting idea. However, for us:
>
> - It is not clear, how a substitution can be optimal or equivalent in effectiveness, particularly in terms of achieving diversity and robustness, which are fundamental goals in our DISCO framework. The primary reason for using all of the models during training is to ensure that the ensemble learns a diverse set of *representations* enforced by Equation (10). If we were to train only a random subset of models at each step, the optimization process would lack cohesion, potentially leading to underutilized model capacity, less diversity, etc.. Further, it would also remove the conditions under which the proof and derivation for Equation 10 is undertaken in [2] and generalized in [3]. So, we expect, this will reduce the effectiveness of the model set in providing robustness against adversarial attacks.
>
> -  It is also not clear how long it would take to converge to an effective solution. Sampling subsets of models during training could undermine the balance between diversity and accuracy, as some models are optimized more effectively than others. Consequently, this approach may converge slower to an effective solution and cost more training time.
>
> [2] Qiang Liu and Dilin Wang. Stein variational gradient descent: A general purpose Bayesian inference algorithm. In Neural Information Processing Systems, 2016.
>
> [3] Dilin Wang and Qiang Liu. Nonlinear stein variational gradient descent for learning diversified mixture models. In Proceedings of the 36th International Conference on Machine Learning (ICML),2019.
>
> *Please kindly reach out, if you have any further questions, we stand ready to answer them promptly to help clarify any further issues.* Once again, we thank you for seeking these clarification and your kind input and feedback.

---

> > ### Comment · Reviewer_YFcN · 2024-12-03
> >
> > Thank you for your clarification. The overhead in training and inference is indeed the main drawback of this work. However, considering the novelty of the approach, I am inclined to accept the paper.
> > I just want to have a small comment. The main question of the paper, "*Can Model Randomization Offer Robustness Against Query-Based Black-Box Attacks?*", I believe has already been answered in previous studies such as RND and RF. This paper proposes a more effective way to utilize that property in defending the model.

---

> ### Author Response · Authors · 2024-12-04
> **Thank you for your constructive feedback. We discuss adding noise to inputs (RND) or  features (RF) vs. model randomization.**
>
> We really appreciate the effort the Reviewer has spent engaging with us and the insightful discussions. We hope ICLR rewards and recognizes this.
>
> We also appreciate the recognition of the novelty and contributions of our work.
>
> We wanted to kindly highlight the distinctions between our approach and prior methods RND and RF. We believe the key question raised in our paper has not been discussed and answered in previous studies and we hope this clarification makes our paper standout.
>
> As we discussed in *Section 2*, Randomized Noise Defense (RND) and Randomized Features (RF) methods explored randomization in input (adding noise to the input) and feature spaces (injecting random noise into computed features). So, it is arguable if this constitutes a randomization of models, as the model represented by the parameter $\theta$ where $f(x,\theta)$ describes the model, is not changed or randomized. So RND and RF do not randomise models, as the model parameters are not altered or somehow different for each query.
>
> In contrast, our method ***does not rely on noise at all***. We actually randomize the model or select a different function or different model parameter $\theta$ for each query. This approach leverages model diversity and response confusion (by randomly selecting models or model randomization). We don't touch inputs or features or outputs. Diversity complements response confusion experienced by the attacker.
>
> Importantly, our approach and its implementation (with learning objectives in Equation 10 & 11) allows the defender to enhance robustness whilst mitigating the performance trade-offs associated with noise-based defenses.
>
> So these are our thoughts. Thank you so much for the discussion!

---

### Official Review · Reviewer_WLVn · 2024-11-04

**Soundness:** 3
**Presentation:** 3
**Contribution:** 3
**Rating:** 5
**Confidence:** 4

**Summary:**

The paper proposes a defense mechanism against query-based black-box attacks relying on model randomization. The proposed method aims to hinder the estimation of the gradients needed to compute adversarial examples obfuscating the relationship of successive queries by randomly sampling a model from a set of models for each query. The paper includes a theoretical analysis showing how this strategy forces the attacker to increase the number of queries to obtain an accurate estimate of the gradient and how model diversity helps to enhance robustness against gradient-free attacks. The experimental evaluation shows that this method is more robust than other strategies relying on randomization for defending against query-based black-box attacks and that the proposed approach to increase models’ diversity helps also to increase robustness.

**Strengths:**

+ The authors propose a novel approach to generate output diversity for defending against query-based attacks relying on randomization. The propose a new strategy to enhance models’ diversity minimizing the impact on the clean accuracy.

+ The authors provide a nice theoretical analysis justifying why model randomization is effective against both gradient estimation and gradient-free attacks. In the first case, Proposition 1 shows that this approach forces the attacker to increase the number of queries to produce accurate gradient estimates and, for gradient-free attacks, Proposition 2 shows how increasing model´s diversity increases the probability of misleading an attack direction, enhancing the robustness to query-based attacks.

+ Compared to other defenses that have a negative impact on the clean accuracy, the proposed defense aims to achieve both high robustness and clean accuracy. For this, the proposed strategies to select models’ subsets and increase diversity show good empirical results.

+ The experimental evaluation includes a good representation of state-of-the-art attacks and defenses against query-based attacks. The results show that the proposed method improves robustness compared to other competing methods relying on randomization and that the mechanism proposed by the authors to increase diversity is important to achieve such goal.

**Weaknesses:**

+ Compared to other randomization-based strategy, training and storing a diverse set of models can be computationally demanding, especially for large models and training datasets. In this sense, the computational complexity is not well discussed. It would be convenient to compare the complexity of this approach with the other competing randomization-based defenses mentioned in the paper, as well as to discuss better the trade-offs between robustness and computational burden.

+ Following up with the previous point, the computational complexity of model sampling and diversity training can have important scalability and latency issues in some practical applications (e.g. real time) or in common scenarios of modern machine learning systems, where the models and the datasets are large. In contrast, for smaller models, other alternatives to query-based strategies, like transfer attacks, can be more appealing to attackers, which can limit the capacity of the proposed approach to defend against attacks. In this sense, I think that the authors should position better the paper and discuss the type of scenarios where such a defense can be useful and applicable.

+ Although the theoretical analysis provides some good insights that support the benefits of the proposed method, the result in Proposition 1 is somewhat limited, as it relates the number of queries to achieve some quality in the gradient estimation as a function of the upper and lower bound for the gradient’s value. Thus, the result does not relate to the characteristics of the random model selection method. For instance, it would be interesting to analyze how the size of the subset K has an impact on the number of queries required to achieve an error estimation in the gradient lower than Delta.

+ Although the empirical endorse the use of equation (10) to train diverse models, the motivation for proposing this approach is not well motivated and justified. Perhaps the authors could provide a more detailed explanation about the motivation and justification.

**Questions:**

+ Computational complexity: Can the authors provide some insights about the computational complexity and the applicability of the proposed defense compared to other competing methods based on model randomization?

+ Can the authors provide more details on the motivation and justification of equation (10) to train more diverse models?

---

> ### Author Response · Authors · 2024-11-23
> **We addressed all of the Weaknesses and Questions in your Feedback with New Results.**
>
> ***Thank you for the initial evaluation of our paper.***
>
> __Q1. Cost Analysis and Complexity Mitigation Strategy__
>
> Indeed the Reviewer is right, there is *no free lunch*.
>
> We achieve much better robustness compared to previous methods. But, model randomization does lead to increasing the training and storage burden. RND and Random Feature use a single model, we employ a set of $n$ models so the number of parameters in our approach is $n \times$ higher and the memory consumption is also larger.
>
> Fortunately, this problem can be mitigated:
>
> - Recent work has begun tackling this problem for ensembling methods. For example, the study [1] shows how a pre-trained model can be trained with less than a ***1%*** increase in parameters and storage cost to build ensembles of diverse models. The authors use CLIP for ImageNet and VBL for language tasks.
> - Importantly, this recent work builds upon research into efficient model tuning with low-rank adapters (LORAs) [2]. Based on these studies, we'll add a section to discuss how to address the issue with training and storage.
>
> Following the recommendation from the reviewer, we report:
>
> **(1)** Cost comparisons for training a single model vs. a set of models (40 models for MNIST, ten models for CIFAR-10/STL-10) used in our experiments to show that achieving better robustness does come at some cost. However, that cost can be mitigated through methods as [1].
>
> Table 1: Training time of models trained on different datasets between a single model and a set of models (DISCO).
> |Datasets|Single Models|DISCO|
> |---|---|---|
> |MNIST|~0.5 hr|~12.5 hrs|
> |CIFAR-10| ~1.5 hr|~72.0 hrs|
> |STL-10| ~1.2 hr|~60.0 hrs|
>
> Table 2: Trainable Parameters of models trained on different datasets between a single model and a set of models (DISCO).
> |Datasets|Single Models|DISCO|
> |---|---|---|
> |MNIST|0.312 M|12.5 M|
> |CIFAR-10|14.73 M|147.3 M|
> |STL-10|11.18 M|111.8 M|
>
> _Table 3:_ Storage Consumption of models trained on different datasets between a single model and a set of models (DISCO).
> |Datasets|Single Models|DISCO|
> |---|---|---|
> |MNIST|1.19 MB|47.7 MB|
> |CIFAR-10|56.18 MB|561.84 MB|
> |STL-10|43.15 MB|426.55 MB|
>
> **(2)** __ImageNet with CLIP (at only 1% training cost )__
>
> - We added new results, following the method in [1], for ImageNet using a set of 5 models and selecting 2 out of 5 for model randomization.
> - Now:
>   - The cost in training is to update just 1.84M parameters instead of 570M for 5 models (notably each CLIP model updates 114M parameters). This is just 0.38% of the cost for training 5 models. So, less than ***1%*** (1.84M/5) of parameters in a single CLIP model needs to be updated to build a *single* model in an ensemble.
>   - The storage for a single CLIP mode is 433 MB. Using the LoRA-based method, our 5 models consume just 439 MB, since parameters not updated during tuning do not need to be replicated. This is only a 1.38% increase in storage for 5 models compared to a single model. So, less than ***0.3%*** increase in storage is required for each model in an ensemble.
> - But we still benefit from the ***improved robustness*** to attacks.
>
> Table 4: $l_\infty$ objective. Robustness ($\uparrow)$ of different defense methods against SQUAREATTACK with the __ImageNet__ task with __OpenCLIP__ architecture [3] (For experimental settings, kindly see the response to __Reviewer YFcN__).
> |Methods|0.025|0.05|0.075|0.1|
> |---|---|---|---|---|
> |RND|83.39%|61.95%|43.37%| 24.89%|
> |RF|86.45%|65.1%|51.14%| 35.83%|
> |DISCO|__90.76%__|__72.51%__|__56.17%__|__45.4%__|
>
> **Q2. Application Dilemma - Better robustness or reduce the cost of training?**
>
> - Our work aims to theoretically investigate an alternative defense *idea* capable of better robustness.
> - Recent research, as we reported, shows that we can mitigate the issues of computational complexity. So, trading-off a small increase in training cost can achieve improved robustness.
> - This is relevant for, model service offerings in applications such as finance, healthcare, or defense where the cost of adversarial failures is significant, and robustness takes paramount.
>
> **Transfer attacks?**
>
> - As we discussed in **Section 2**, transfer-based attacks’ success relies on the similarity between the surrogates and target models. [4, 5], & [6] pointed out that the success of transfer attacks is limited for diverse models. Therefore, if an adversary favors transfer attacks, our approach is well-suited for these attacks. _Simply, the diverse models help reduce the similarity between the defended model and the surrogate model used by an adversary_.

---

> ### Author Response · Authors · 2024-11-23
>
> __Q3. An analysis of the impact of the size of the subset of models on the number of queries.__
>
> Thank you for your thoughtful question. Proposition 1 relates directly to the number of queries but doesn't explicitly link to the random selection method. The link follows naturally from Proposition 1, and we are sorry we didn't make this clear.
>
> Following your suggestion, we can provide an additional analysis for the trade-off between the selection of $N$ (the subset set size) from $K$ models and the number of queries to achieve a low error estimation.
> - Intuitively, a larger subset size $N$ reduces the _number_ of combinations of model subsets. This results in a reduction in the number of random models presented to the attacker.
> - In addition, a larger subset size $N$ also reduces the variance in estimates of gradient attempted by an attacker. Because, the averaged prediction from a larger set of models is more confident and the variance, for example, in output scores between these large subsets is less.
> - Consequently, averaging across larger subsets of models leads to more informative resoponses (better gradient estimates for example) and less queries to obtain low error estimations.
> - In contrast, smaller $K$ values increase the uncertainty, which leads to increased variance in gradient estimation or in other words the difference in upper and lower bound for the gradient’s value will be larger. Then following ***Proposition 1***, this increase the cost of the attack, that force the attacker to expend more queries to obtain a low error estimation of a gradient.
>
> __Q4. Motivation for Equation (10) is to Encourage Parameter Diversity__
>
> We presented our motivation and justification in **Section 3.4.1**. We can clarify further here. To explain:
> - Recall, our hypothesis (see Hypothesis 2) was "_Randomly sampling functions or models from a set with very diverse parameters should increase diversity in outputs_"
> - The objective in (10) effectively pushes model parameters apart (in the Bayesian context to better approximate a multi-modal posterior). This forces learning different and diverse represenations as shown in prior work [7] to yeild, effectively, a set of functions with different parameters.
> - Then we can expect sampled functions (models from such a set of models) to result in output diversity. Consequently, leading to high output variance.
> - Importantly, our diversity analysis results in ***Figure 3*** can demonstrates and confirm our motivation is justified.

---

> ### Author Response · Authors · 2024-11-24
> **Answers to Questions**
>
> - Computational complexity?
>
> *Response: Please refer to our answer for **Question 1** above in addressing the Weaknesses in the paper.*
>
> - Can the authors provide more details on the motivation and justification of equation (10) to train more diverse models?
>
> *Response: Please refer to our answer for **Question 4** above in addressing the Weaknesses in the paper.*

---

> ### Author Response · Authors · 2024-11-25
> **References**
>
> [1] Bao Gia, D, Shamsi, A, Guo, X-Y, Mohammadi, A, Alinejad-Rokny, H., Sejdinovic, D, Ranasinghe DC, & Abbasnejad, E. Bayesian Low-Rank LeArning (Bella): A Practical Approach to Bayesian Neural Networks. ArXiv (2024)
>
> [2] Hu, EJ.; yelong shen; Wallis, P.; Allen-Zhu, Z.; Li, Y.; Wang, S.; Wang, L.; and Chen, W. LoRA: Low-Rank Adaptation of Large Language Models. ICLR, 2022.
>
> [3] Radford, A., Kim, JW., Hallacy, C., Ramesh, A., Goh, G., Agarwal, S., Sastry, G., Askell, A., Mishkin, P., Clark, J., Krueger, G., & Sutskever, I. “Learning Transferable Visual Models From Natural Language Supervision.” ICLR, 2021.
>
> [4] F. Suya, A. Suri, T. Zhang, J. Hong, Y. Tian, & D. Evans. Sok: Pitfalls in evaluating black-box attacks. (SaTML), 2024.
>
> [5] Pin-Yu, C., Zhang, H., Sharma, Y., Yi, J., & Hsieh. C-H., Zoo: Zeroth order optimization based black-box attacks to deep neural networks without training substitute models. AISec, 2017.
>
> [6] Cheng, S., Miao, Y., Dong, Y., Yang, X., Gao, X., & Zhu, J. (2024). Efficient Black-box Adversarial Attacks via Bayesian Optimization Guided by a Function Prior. ICML, 2024.

---

> > ### Comment · Reviewer_WLVn · 2024-11-26
> > **Response to the authors' comments**
> >
> > Thank you very much for your detailed response and your effort in providing additional results.
> >
> > Indeed, there is no free lunch, and there are important trade-offs between the models’ accuracy, robustness, and computational complexity that need to be considered. While I appreciate the authors’ effort in providing some additional information about the overhead of the proposed approach, I believe that the experiments should better reflect these trade-offs to have a more comprehensive overview of when DISCO can be an appealing defense to use, regarding the model’s size, the training set, or the number of models to be trained. For this, I think the experiments must be revised and reconsidered, and I do not think that a minor revision of them will suffice.
> >
> > The computational complexity has been an argument raised by most reviewers and I think that the authors could make a more compelling case for defending their approach. For instance, the directions pointed out in their response to my Q1 can be promising and worth exploring in more depth. It would be interesting to analyze whether LoRA can produce diverse models and reduce the computational complexity for DISCO. On the other hand, as mentioned in my review, I think that the authors can also do a better work positioning the paper by, for example, analyzing in more depth applications or scenarios where DISCO can be an appealing defense and where the extra memory and training time can be justified for gaining robustness, compared to other defenses.
> >
> > I believe that, despite the limitations with the computational complexity, the paper has potential, but the changes that need to be addressed are not minor. I appreciate some of the extra result reported during the rebuttal, like the analysis of ImageNet and the evaluation against the P-BO attack are interesting and show promising results. However, as mentioned before, I think that the experiments should be reconsidered to make a more compelling case that justifies the extra complexity and reflects better the trade-offs between accuracy, robustness, and complexity. For these reasons I am keeping my score, but I really encourage the authors to keep on working on this defense and improve the paper.

---

> ### Author Response · Authors · 2024-11-27
> **We have answered all of the new questions (ImageNet on CLIP cost analysis show cost is < 0.32% per model, results in Appendices F &G, new Section 4.5 in updated paper)**
>
> **Q2** *better reflect these trade-offs*
>
> We certainly appreciate the Reviewer's sentiment that the main body of the paper should make the cost clearer to the reader, so the performance gains are put in perspective.
> - Certainly, in the paper we clearly mention the number of models we used in table captions and text.
> - Then we report sampling from a fixed number in the main body due to the overwhelming nature of the results set for other strategies, defer these to the ***Appendices F & G***
>
> Now we will explicitly discuss the costs trade-off and add this to the main body of the paper as a new ***Section 4.5***. We absolutely want to allow the research community to benefit from our in-depth analysis and results and be very upfront about the fact that there is no-free-lunch.
>
> - We will share the updated PDF with you to show this addition and the cost analysis recommended by the reviewer.
>
> **Q2** *The computational complexity has been an argument raised by most reviewers and I think that the authors could make a more compelling case for defending their approach.*
>
> *Response:*
>
> We believe we have a very compelling case.
>
> We start with a***pre-trained CLIP*** model to demonstrate practicability and build a working defense for a model suitable for serious applications.
>
> - We stated with a ***pre-trained CLIP*** (to put CLIP into perspective, please note that ***RestNet50 has approximately 25 million parameters compared to 114 million we deal with in CLIP***).
> - We then trained an ensemble of 5 using the methods in [3].
>     -  The model set achieves approximately 78% clean accuracy.
>     -  During the inference phase, two out of five were randomly selected from the trained set.
>     -  Two out of five has a clean test accuracy of approximately 77%
> - We also train a single CLIP model for use with RND and RF
>     - The single model achieves 76.07% clean accuracy
>     - We add noise, so the clean accuracy drop values are as in RF (ICLR'24), approximately 1% and comparable to ours.
> - Table 4 shows that the significant reduction in overhead we achieve.
> - Table 5 shows that compared to current state-of-the-art we outperform with a margin of up to 9.57%
>
> _Table 4:_ Trainable Parameters and Storage Consumption of a Single CLIP and a set of five CLIP models ***we trained*** to implement LoRA(DISCO).
> | Models|***Single*** CLIP|The set of 5 CLIP models using LoRA|
> | -------------------- | -----------|------|
> | Trainable Parameters |114 M| 1.84  M  (1.6% incrase, ***0.32% per model***)|
> | Storage Consumption  |433 MB|  439 MB (1.35% increase, ***0.28% per model***)|
>
> _Table 5:_ $l_\infty$ objective. Robustness ($\uparrow)$ of different defense methods against SQUAREATTACK with the __ImageNet__ task task with __CLIP__ model architecture [3] (For details on the experiment, please see response to Reviewer).
> | Methods | 0.025      | 0.05       | 0.075      | 0.1       |
> | ------- | ---------- | ---------- | ---------- | --------- |
> | RND     | 83.39%     | 61.95%     | 43.37%     | 24.89%    |
> | RF      | 86.45%     | 65.1%      | 51.14%     | 35.83%    |
> | DISCO   | __90.76%__ | __72.51%__ | __56.17%__ | __45.4%__ |
> | DISCO Improvement (vs. Next best)|4.31%|7.41%|5.03%|9.57%|
>
> What we are proposing are ***marginal cost increases*** to achieve significant improvements in robustness.
>
> - Effectively we are saying <1.6% increase in overhead can yield 4.31 to 9.57% better robustness on a large-scale network of practical significance
> - Now adding a model incurs **<0.32%** overhead in terms or trainable parameters or storage.
>
>
> We really hope the results provides the assurances sought by the Reviewer that our methods is:
> - Robust and
> - Practical
>
> We hope the reviewer can appreciate the question we posed in the paper:
> - CAN MODEL RANDOMIZATION OFFER ROBUSTNESS AGAINST QUERY-BASED BLACK-BOX ATTACKS?
>   - We believe the answer is now, yes, irrevocably.
>   - In addition to understating what such a method can offer, and the theoretical analysis, we have now shown it can be of practical significance.
>   - One paper can't solve all problems, but we have certainly worked hard at making our theoretical work stick.
>
> ***Notably, consider using adversarial training to make a model like CLIP more robust with be an insurmountable cost. Ours is a relatively simple method we can perform even on an A6000 GPU.***
>
> We sincerely thank the Reviewer for all their efforts to help us improve our work and its presentation. Please let us know if there are any specific results or experiments the Reviewer would like to see. We stand ready to provide this for you.

---

### Author Response · Authors · 2024-11-28
**We updated the paper (Added new results recommended by Reviewers, Provided a cost analysis, With new results for practicability Using OpenCLIP)**

We just wanted let you know that we have updated the paper.

 - We edited and added content to reflect all of your comments (***New Appendices C, E F and G***)
 - We have a comprehensive cost analysis (***Section 4.5***)
 - We have addressed the key concern around the cost of implementing on a large network and high-res dataset and application relevance by using ***ImageNet*** on ***OpenCLIP*** (114 million parameters, open-source implementation of ***OpenAI's CLIP***).
- We have added a cost analysis for implementing our method with OpenCLIP (**<0.32%** storage and training cost increase per model)
 - We have added new attack results for ***PB-O*** (a much stronger attack under a surrogate model-based setting).
- Then compared to ***RF (ICLR'24)***, current SToA, our method is a *new* idea, is more robust, even on ImageNet - we set a **new benchmark result for a defense**.

We also want to kindly highlight:

*In all our extensive experiments, including the new and strong PB-O attack, our **new method** sets a new benchmark compared to the baseline RF (ICLR'24) and RND (NeurIPS'21 & CVPR'22), with significant margins (up to 9.5% with ImageNet).*

*Our new method, as carefully reviewed by all 6 Reviewers, is supported by our **theoretical analysis**. Further the method is practical to implement, even with a large-scale network like OpenCLIP and high-resolution datasets like ImageNet.*

*Most importantly, in pursuing our insights into the method to confuse attackers, it was never immediately clear from existing literature, if model randomization can lead to sufficient obfuscation to confuse query-based black-box attacks or how best to build such a method.* Our work:

- Shows model randomization can obfuscate the relations exploited by attackers in back-box settings.
- Our learning objectives (Equation 10 and proposed Equation 11) provides an effective means to implement the new idea (In fact Equation 10 and 11 outperform other methods in the literatures we tried).


Thank you very much for all the constructive discussions.

---

### Note · Authors · 2025-02-23

I have read and agree with the venue's withdrawal policy on behalf of myself and my co-authors.

---

### Meta-Review · Area_Chair_5QzV · 2024-12-17

**Metareview:**

The reviewers were conflicted about accepting this paper, and even the positive reviewers (none of which gave higher than 6) were skeptical that this defense could be practical, especially given the high computational costs.  Moreover, while I appreciate the discussion in the rebuttal period, addressing all reviewer concerns requires a massive change to the original submission which may be best left for a future submission.  Therefore, I recommend rejection for this paper, but encourage the authors to keep improving their work.

**Additional Comments On Reviewer Discussion:**

The authors engaged heavily with reviewers during the rebuttal period, including numerous new experiments.  Nonetheless, reviewers were still lukewarm at the end of the period, and nobody chose to champion this paper.

---

### Decision · Program_Chairs · 2025-01-22

Reject